# Antibiotic stewardship benchmarking–Using the WHO point prevalence survey of antimicrobial prescribing in a Tertiary Care Public Hospital, Karachi

Ale Zehra[1,◉*], Tehreem Ansari[2◉], Syed Shaukat Ali Muttaqi Shah◉[3◉*], Beenish Syed[2‡], Mehwish Rizvi[4‡], Fakhsheena Anjum[1‡], Tehrim Fatima[5‡], Farah Saeed[6‡]

**1** Department of Pharmacy Practice, Faculty of Pharmaceutical Sciences, Dow College of Pharmacy, Dow University of Health Sciences, Karachi, Sindh, Pakistan, **2** Department of Infectious Disease, Dow University Hospital (Ojha Campus), Dow University of Health Sciences, Karachi, Sindh, Pakistan, **3** College of Pharmacy, Al-Farahidi University, Baghdad, Iraq, **4** Department of Pharmaceutics, Faculty of Pharmaceutical Sciences, Dow College of Pharmacy, Dow University of Health Sciences, Karachi, Sindh, Pakistan, **5** Department of Pharmacology, Faculty of Pharmaceutical Sciences, Dow College of Pharmacy, Dow University of Health Sciences, Karachi, Sindh, Pakistan, **6** Department of Pharmacognosy, Faculty of Pharmaceutical Sciences, Dow College of Pharmacy, Dow University of Health Sciences, Karachi, Sindh, Pakistan

◉ These authors contributed equally to the work.
‡ These authors also contributed equally to the work.
* ale.zehra@duhs.edu.pk (AZ); syedshaukatali@uoalfarahidi.edu.iq (SSAMS)

## Abstract

### Background

Antimicrobial resistance (AMR) is a global threat, mainly linked to inappropriate use and prescription of antibiotics, Antimicrobial stewardship (AMS) programs have proved to promote responsible antibiotic use and decrease the burden of AMR. The aim of this study is to benchmark antibiotic prescribing patterns and evaluate stewardship practices using the World Health Organization (WHO) Point Prevalence Survey (PPS) methodology in a tertiary care public sector hospital in Karachi.

### Method

A cross-sectional, prospective PPS was conducted over four weeks in July 2024 at Dow University Hospital, Karachi. The data were extracted from the medical records of the patients using a validated WHO PPS tool by a trained infectious disease physician and pharmacist. All inpatients admitted before or at 8:00 a.m. on survey day, without a planned discharge were included, excluding those from emergency, acute care, day-care surgery, dialysis, and oncology units. Descriptive analysis of the data was performed using Stata version 14.

**Data availability statement:** All relevant data are within the paper and its Supporting Information files.

**Funding:** This research was funded by the Dow University of Health Sciences, Vice Chancellor Seed Funding Project 2023–2024 (Ref: DUHS/VC/2023/11-04/01). The funder provided financial support for data collection and statistical analysis but had no role in study design, interpretation of data, decision to publish, or preparation of the manuscript.

**Competing interests:** The authors have declared that no competing interests exist.

**Abbreviations:** AMR, Antimicrobial Resistance; AMS, Antimicrobial stewardship; ASP, Antimicrobial Stewardship Program; ATC, Anatomical Therapeutic Chemical; CAI, Community Acquired infection; CDC, Centers for Disease Control and Prevention; CSF, Cerebrospinal Fluid; DTC, Drug Therapeutic Committee; ER, Emergency Room; GAP, Global Action Plan; HAI, Hospital Acquired infection; HDU, High Dependency Unit; ID, Infectious Diseases; ICU, Intensive care unit; IQR, Interquartile Range; IV, Intravenous; LMICs, Low and Middle Income Countries; MP, Medical Prophylaxis; PPS, Point Prevalence Survey; SD, Standard Deviation; SP, Surgical Prophylaxis.

## Results

Out of 224 hospitalized patients at the day of survey, 186 inpatients (adults and children across medical, surgical and critical care wards) were included in the study meeting the inclusion criteria. The study included 50.5% male and 49.5% females, having mean age of 45 (±18) years. The point prevalence of antibiotic use was 83.3% (95% CI: 77.5–88.2%). Community-acquired infections 55.5% (95% CI: 48.7–62.1%) were the most common indication of use. Most antibiotics 99.2%, (95% CI: 95.6–99.9%) were prescribed empirically, with predominant parenteral administration 89.2% (95% CI: 84.5–92.9%) and limited Intravenous-to-oral switch 2.9% (95% CI: 1.3–6.2%). Ceftriaxone (18.5%), piperacillin-tazobactam (18.1%), and meropenem (16.2%) were most frequently used antibiotics. According to WHO Access, Watch and Reserve (AWaRe) classification, 80.8% (95% CI: 75.2–85.6%) of antibiotics belonged to the 'Watch' category, 17.3% (95% CI: 12.6–23.2%) to 'Access', and 1.8% (95% CI: 0.7–4.6%) to 'Reserve'. Cultures showed no growth in 64.8% (95% CI: 55.2–73.6%) of cases. Stewardship interventions were found applicable in 55.4% (95% CI: 48.7–62.0%) of prescriptions due to overuse, dosing errors, and absence of antimicrobial guideline in the hospitals.

## Conclusion

This study demonstrates that antibiotic utilization exceeded global averages, highlighting the urgent need to develop institutional antimicrobial guidelines, enhance stewardship programs, and improve diagnostic stewardship to curb AMR.

## Introduction

Antibiotics are one of the most often recommended drugs for both prophylactic and therapeutic purposes in hospital settings. They were discovered as the magic bullet to treat infectious disorders for the safety of humanity [1,2]. But unfortunately, their benefits are limited to their rational use only and inappropriate administration results in negative consequences and adverse health outcomes [3]. Irrational prescribing or administration patterns may result in worsening patient outcomes, increase healthcare costs, and most of all Antimicrobial Resistance (AMR) [4].

Antibiotic resistance is the self-defense mechanism used by microorganisms against antibiotics [5]. Resistance is a natural occurrence that allows organisms to adapt and alter, making medications futile [6]. AMR is a significant clinical and economic burden of this era, one that clinicians are trying to address. It poses a serious risk to public health worldwide [7–9]. The rate of AMR has risen significantly due to inappropriate antibiotic prescribing patterns [10–12]. It has become the need of the hour to take control of the rapidly growing AMR to preserve the effectiveness of existing antibiotics against numerous microorganisms and to treat multiple ailments. Monitoring current patterns of use and launching initiatives to rationalize the use of antibiotics when necessary through Drug and Therapeutics Committees (DTCs) and

antimicrobial stewardship programs (ASPs) is a crucial first step in enhancing the use of antibiotics in hospitals [13–15]. ASPs is defined as a set of systemic and coordinated interventions that aim to optimize the use of antimicrobial agents in a setting [16]. These interventions improve rational antibiotics use by, measuring the use of antibiotics, promoting proper selection of appropriate antibiotics regimen, including suitable dose, duration and preferred route of administration, without compromising patient outcome [17,18]. This rational use can be achieved by keeping track of how often antibiotics are prescribed in comparison to established essential drug lists or standard guidelines [19,20].

The World Health Organization (WHO) suggests the implementation of ASPs to identify and monitor antibiotic usage and scale down the burden of AMR in an effort to mitigate the risks posed by it [21]. Healthcare providers are now recommended to limit the prescribing of antibiotics for rational use only, to maintain their effectiveness against various microbes [22,23]. Appropriate dosing parameters should be followed for an optimal duration following the standard protocols to minimize AMR and reduce the resulting morbidity, mortality, and financial cost.

The rapidly growing AMR around the world has now reached an alarmingly high level, due to the irrational use of various broad-spectrum antibiotics in clinical settings that were solely reserved for some specific disease conditions [24,25]. The research to date indicates that 20–50% of the antibiotic prescription pattern is irrational [26]. WHO established a Global Action Plan (GAP) in response to rising issues about AMR and its consequences [27]. An important objective of GAP is to provide measures to ensure the quality use of antibiotics, hence lowering future rates of inappropriate antibiotic use and AMR [28,29]. One tactic to accomplish these objectives is to regularly monitor the use of antimicrobials by conducting point prevalence surveys (PPS). A PPS is a standardized method used in healthcare settings to assess microbial use and prescribing patterns at a specific point in time [30,31]. Its primary goal is to observe prescribing patterns and compare them with the established guidelines of WHO or Centers for Disease Control and Prevention (CDC). PPS is frequently used to establish benchmarks for practice and evaluate effectiveness of AMS programs [32]. Following these guidelines, multiple PPS are being carried out globally but a limited number have been reported from Pakistan, primarily in private or other regional hospital settings. The PPS of antimicrobial use will be helpful to analyze the antibiotic administration pattern and its compliance with standard WHO or CDC guidelines. It is frequently used to pinpoint areas for improvement initiatives, establish benchmarks, and evaluate the effectiveness of antimicrobial stewardship measures [33]. The data from major public tertiary care hospitals in Karachi specifically remains sparse, limiting the generalizability of national stewardship benchmarks. This study therefore aims to apply WHO-PPS methodology in a major public sector hospital in Karachi. By providing a detailed, benchmarked snapshot of antibiotic prescribing and stewardship gaps in a high patient-volume setting, the study seeks to fill a significant gap and generate locally relevant data to inform the development of tailored stewardship interventions, institutional guidelines, and diagnostic improvements, necessary to curb AMR in the public healthcare system. This study therefore contributes to the limited but growing body of WHO-PPS data from Pakistan by providing a detailed snapshot of antibiotic use in a public tertiary care hospital in Karachi, a setting underrepresented in the national AMS literature.

## Materials and methods

A cross-sectional, prospective study was conducted over four weeks (1–30 July 2024) in various wards of the Dow University Hospital, Karachi. Data were collected using the standard patient case report tool "S1 Appendix" and an antimicrobial use form "S2 Appendix" that were adapted from the PPS technique provided by the latest version (1.1) of the WHO antibiotic methodology, published in 2019 [6]. A trained Infectious Diseases (ID) physician and a pharmacist collected data from patient files/ medical records. Ethical approval was obtained from the Institutional Review Board of the Dow University of Health Sciences, Karachi, reference number (IRB-3407/DUHS/Approval/2024/102), dated: 28th March 2024. The data collection was conducted in accordance with ethical principles to ensure participant confidentiality, anonymity, and compliance with the Declaration of Helsinki [34,35]. A written informed consent was obtained from all the participants in English and Urdu (local language). For participants unable to provide consent, written consent was obtained from their legal guardians and documented.

All the inpatients at the hospital, admitted before or at 8:00 am on the day of survey of the ward, were included in the study. Each ward was completely surveyed within one day to minimize the effect of patient transfers between wards following the WHO PPS methodology. In accordance with the WHO PPS protocol, patients who were temporarily absent from the ward (e.g., for medical imaging, endoscopy, or surgery) were included only if they were admitted to the ward and expected to return later on the survey day. Patients from emergency, dialysis, chemotherapy, or day-care surgery units were excluded.

Patients receiving antimicrobials were enrolled if they were on at least one antibiotic at 08.00am on the survey day, patients whose antibiotics were started after 08:00am or discontinued before 08:00am were excluded. Information regarding wards, including ward name, type of ward as per WHO, total number of patients, number of eligible patients, and number of included patients on the day of the survey, was noted. Patient-related variables included demographic details (age, gender, admission information, diagnosis, comorbidities including HIV status), date of procedure (if surgery occurred between admission and the date of the survey), and prior hospital exposure within 90 days, and referrals from other hospitals. Risk factors increasing the likelihood of antibiotic use, such as the presence of catheters or central lines, were recorded in "S1 Appendix". Antibiotic related information including the number of antibiotics prescribed, indication (surgical prophylaxis, medical prophylaxis, hospital-acquired infection, or community-acquired infection), type of treatment (empirical, targeted, pre-emptive), site and type of infection, dose, frequency, duration, route, and date of initiation was recorded in "S2 Appendix". In cases of surgical prophylaxis, dose, duration, and site of procedure were also documented. Antibiotics were classified according to the WHO AWaRe (Access, Watch, Reserve) categorization system. The AWaRe classification is a WHO tool to guide antibiotic selection and stewardship by grouping antibiotics based on their impact on antimicrobial resistance [36,37]. The 'Access' group includes first-line (Access = first-line), narrow spectrum agents; the 'Watch' group comprises antibiotics with higher resistance potential and should be monitored (Watch = monitored); and the 'Reserve' group contains last-resort options (Reserve = last-resort).

Microbiological variables such as date and site of specimen collection, identified organisms, and resistance patterns were recorded in "S3 Appendix". The prescribed antibiotics were also checked for stewardship assessment using "S4 Appendix"; if applicable, stewardship alerts were noted. Stewardship opportunities were identified and categorized using criteria adapted from standard ASP principles [37]. Categories included: over prescribing (antibiotic use without clear clinical indication); incorrect dose (deviations from weight-based or renal-adjusted doses); overlapping spectrum (redundant antimicrobial coverage); inappropriate antibiotic choice (antibiotic not aligned with cultures or protocols); missed narrow-spectrum opportunity (use of broad-spectrum antibiotic when narrow alternatives are suitable); unnecessary escalation/addition (addition of another antibiotic without documented clinical worsening); extended surgical prophylaxis (SP beyond 24 hours without documented justification); drug-bug mismatch (antibiotic ineffective against cultured pathogen); missed IV-to-oral switch opportunity (eligible patient receiving IV therapy beyond 48–72 hours); use of restricted antibiotic (without ID approval); and microorganism resistant to antibiotic used (based on susceptibility results). The complete list of stewardship criteria in provided in "S4 Appendix".

Medical records of patients who did not receive any antimicrobials were reviewed only for demographic details and not included in stewardship analysis. During the data collection process, no modifications were made in the treatment of patients so the standard patient care was not modified or compromised during the data collection. All data extracted from patient records were entered into a structured spreadsheet. Data validation checks were performed by a second researcher on a 10% random sample to ensure accuracy and completeness. Missing data points, e.g., undocumented antibiotic stop date, were reconciled by re-reviewing the primary medical record, and were recorded as 'unknown' and excluded from analysis to maintain data integrity.

Collected clean data was imported into Stata version 14 (StataCorp LP, College Station, TX, USA) for analysis. Descriptive statistics were used to summarize the data: frequencies and percentages are reported for categorical variables (e.g., indication of therapy, AWaRe category). Continuous variables (e.g., age, duration of therapy) are presented as

                                                                  

mean±Standard Deviation (SD) if normally distributed, or as median with interquartile range (IQR) if the distribution was skewed. For key proportions (e.g., prevalence of antibiotic use, AWaRe category distribution, culture yield), 95% confidence intervals (CIs) were calculated using the Wilson score method to account for binomial uncertainty. No comparative inferential statistics were employed, as the study design was purely a descriptive point prevalence survey. The summarized methodology and data collection is illustrated in Fig 1.

## Results

A total of 224 patients were included in the sampling frame, of which 186 patients met the inclusion criteria of the study. The demographic and clinical characteristics of the study population are summarized in "Table 1".

### Patient demographics and clinical characteristics

The cohort comprised of 94 (50.5%) males and 92 (49.5%) of females, with mean age of 42.3 (SD=±21.6) years. The age distribution included young patients 47 (25.3%, 95% CI: 19.3–32.1%), middle age 46 (24.7%, 95% CI: 18.8–31.6%) and elderly patients 43 (23.1%, 95% CI: 17.4–29.8%), the detailed distribution is given in "Table A in S5 Tables". Most of the patients were admitted to adult High risk ward (44, 23.7%), followed by adult Intensive Care Unit (ICU) 35 (18.8%), adult medical wards 32 (17.2%), adult surgical wards 9 (4.8%), mixed wards 46 (24.7%), Neonatal ICU 5 (2.7%), Pediatrics high-dependency unit (HDU): 2 (1.1%), Pediatric ICU 3 (1.6%), and pediatric medical ward 10 (5.4%), respectively, as elaborated in Fig 1. A total of 131 (70.4%, 95% CI: 63.4–76.7%) were with a single diagnosis, 37 (19.9%, 95% CI: 14.5–26.3%) with two diagnoses and 15 (8.1%, 95% CI: 4.6–12.9%), with three diagnosis and 3 (1.6%, 95% CI: 0.3–4.6%) with more than three diagnosis shown in "Table A in S5 Tables". Most patients 176 (94.6%, 95% CI: 90.4–97.4%) were direct admissions to the hospital, and the majority 172 (92.5%, 95% CI: 87.9–95.8%) had not been hospitalized in the 90 days preceding the current admission. The majority of patients were admitted in the medical ward, 25 Medical ICU (ICUMED, 13.4%, 95% CI: 9.0–19.1%), Medical Gastroenterology (MEDGAST) 19 (10.2%, 95% CI: 6.3–15.4%), Medical General (MEDGEN) 27 (14.5%, 95% CI: 9.8–20.4%), Medical Hematology (MEDHEMA) 15 (8.1%, 95% CI: 4.6–12.9%), Medical Nephrology (MEDNEPH) 15 (8.1%, 95% CI: 4.6–12.9%) and Pediatric General (PEDGEN) 10 (5.4%, 955 CI: 2.6–9.7%), while 19 (10.2%, 95% CI: 6.3–15.4%) were admitted to different surgical specialties as shown in "Table A in S5 Tables". The median length of stay until the survey date was 3 days (range: 1–45), median Total Leukocyte count (TLC) at the time of admission was 9.8 x 10⁹/L (range: 1–96), median creatinine 0.92 mg/dL (range 1–9). A total of 62 (33.3%, 95% CI: 26.7–40.6%) patients were febrile at the time of admission, and only 5 (2.7%, 95% CI: 1.2–6.2%) patients had documented use of antibiotics before admission, while 72 (38.7%, 95% CI: 31.7–46.1%) patients were administered antibiotic in the Emergency room (ER). A total of 163 (87.6%, 95% CI: 82.1–92.0%) patients had peripheral vascular catheters, 94 (50.5%, 95% CI: 43.2–57.9%) had urinary catheter, followed by 8 (4.3%, 95% CI: 1.9–8.3%) with central vascular catheter and 5 (2.7%, 95% CI: 0.9–6.2%) with endotracheal tube.

### Point prevalence survey (PPS) ward assessment

The characteristics of the wards assessed in the point prevalence survey showed on an average, each ward contained 13.7 (±5.2) beds. At the time of the survey (8:00 A.M.), there was an average of 8.4 (±3.7) patients present per ward. Of these, 6.6 (±3.86) patients met the eligibility criteria for inclusion in the survey. There were 1.75 (±2.25) non-eligible patients per ward, mostly because they were admitted after 8:00AM at the day of survey as shown in "Table B in S5 Tables".

### Prevalence and pattern of antimicrobial use

The indications, routes of administration, and patterns of antimicrobial prescribing are elaborated in "Table 2". Of the 186 eligible patients surveyed, 155 (83.3%, 95% CI: 77.5–88.2%) were receiving at least one antimicrobial agent on the day of the survey. Among all recorded indications (n = 218), community-acquired infections, CAI 121 (55.5%, 95%

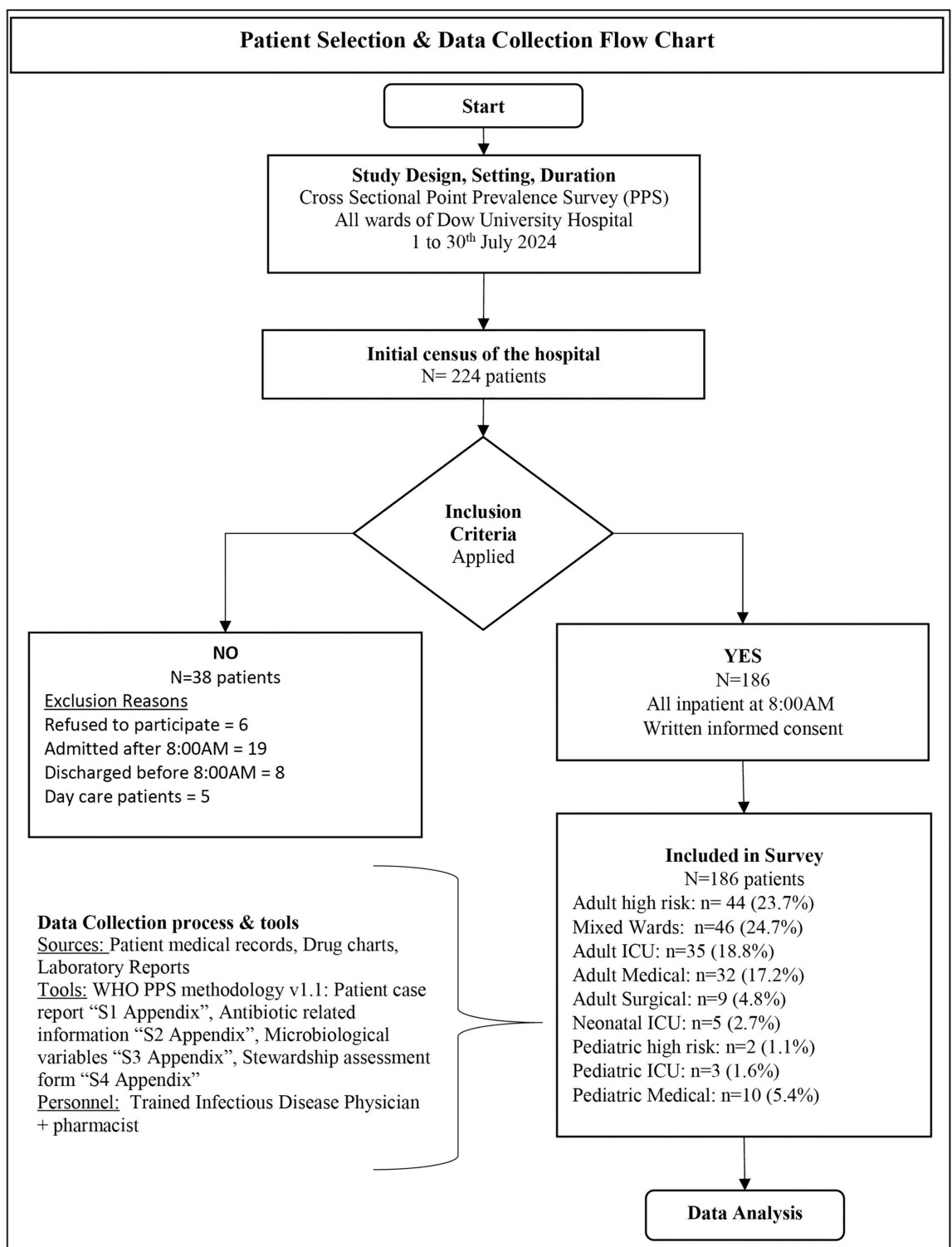

**Fig 1. Flowchart of patient selection and data collection for the point prevalence survey.** The chart details the inclusion and exclusion of patients in the antimicrobial use survey conducted at Dow University Hospital. The final distribution of the 186 included patients across ward specialties is shown. Data were collected by a trained infectious disease physician and pharmacist using WHO PPS methodology v1.1 and standardized case report forms (available in S1, S2, S3, and S4 Appendices).

**Table 1. Demographic and clinical details of patients at admission.**

| Variables | N = 186 | % | 95% CI |
|---|---|---|---|
| **Gender** | | | |
| Male | 94 | 50.5 | 43.2-57.9 |
| Female | 92 | 49.5 | 42.1-56.8 |
| **Age** (years) Mean (±SD) | 42.3 | ±21.6 | |
| **Transfer from other hospital** | | | |
| Yes | 6 | 3.2 | 1.5-6.9 |
| No – Direct admission | 176 | 94.6 | 90.4-97.4 |
| Unknown | 4 | 2.2 | 0.6-5.4 |
| **Previous hospitalization within 90 days** | | | |
| Yes | 7 | 3.8 | 1.8-7.7 |
| No | 172 | 92.5 | 90.4-97.4 |
| Unknown | 7 | 3.8 | 0.6-5.4 |
| **Fever at the time of admission** | | | |
| Yes | 62 | 33.3 | 26.7-40.6 |
| No | 124 | 66.7 | 59.4-73.3 |
| **History of allergy documented** | | | |
| Yes | 48 | 25.8 | 19.8-32.7 |
| No | 138 | 74.2 | 67.3-80.2 |
| **Use of antimicrobial within 90 days** | | | |
| Yes | 5 | 2.7 | 1.2-6.2 |
| No | 56 | 30.1 | 23.7-37.2 |
| Unknown | 125 | 67.2 | 60.0-73.8 |
| **Antibiotic administered in Emergency Room (ER)** | | | |
| Yes | 72 | 38.7 | 31.7-46.1 |
| No | 46 | 24.7 | 18.7-31.6 |
| Not applicable | 68 | 36.6 | 29.7-43.9 |
| **Length of Stay till PPS day,** Median (Min – Max) | 3 | 1–45 | |
| **TLC count at admission,** Median (Min – Max) x 10$^9$/L | 9.8 | 1–96 | |
| **Surgery since admission** | | | |
| Yes | 30 | 16.1 | 11.2-22.2 |
| No | 156 | 83.9 | 77.8-88.8 |
| **Central Venous Catheter** | | | |
| Yes | 8 | 4.3 | 2.2-8.3 |
| No | 178 | 95.7 | 91.7-98.1 |
| **Peripheral vascular Catheter** | | | |
| Yes | 163 | 87.6 | 82.1-92.0 |
| No | 23 | 12.4 | 8.0-17.9 |
| **Endotracheal tube** | | | |
| Yes | 5 | 2.7 | 0.9-6.1 |
| No | 181 | 97.3 | 93.9-99.1 |
| **Urinary Catheter** | | | |
| Yes | 94 | 50.5 | 43.2-57.9 |
| No | 92 | 49.5 | 42.1-56.8 |

Data are presented as n (%) unless otherwise specified. 95% confidence intervals (CIs) for proportions were calculated using the Wilson score method. SD: Standard Deviation; TLC: Total Leukocyte Count; ER: Emergency Room.

**Table 2. Antimicrobial use and indication.**

| | N | % | 95% CI |
|---|---|---|---|
| **Number of patients prescribed antimicrobials** | | | |
| Yes | 155 | 83.3 | 77.5-88.2 |
| No | 31 | 16.7 | 11.8-22.5 |
| **Indication type** | 218 | | |
| Community acquired infection (CAI) | 121 | 55.5 | 48.7-62.1 |
| Medical prophylaxis (MP) | 63 | 28.9 | 23.1-35.5 |
| Surgical prophylaxis (SP) | 20 | 9.2 | 6.0-13.7 |
| Hospital acquired infection (HAI) | 6 | 2.8 | 1.3-6.0 |
| Others (O) | 1 | 0.5 | 0.0-2.6 |
| Unknown indication (UI) | 7 | 3.2 | 1.3-6.5 |
| **Type Of Therapy (CAI or HAI)** | 127 | | |
| Empirical | 126 | 99.2 | 95.6-99.9 |
| Targeted | 1 | 0.8 | 0.0-2.6 |
| **Surgical prophylaxis** | 20 | | |
| SP1(One dose for surgical prophylaxis) | 6 | 30 | 13.2-56.0 |
| SP2(Multiple doses on one day) | 10 | 50 | 27.2-72.8 |
| SP3(Multiple doses for more than 24 hours | 1 | 5 | 0.3-24.7 |
| SPO(if patient hasn't been operated yet but receives AM for SP) | 3 | 15 | 3.2-37.9 |
| **Route of administration of antimicrobial** | 222 | | |
| Oral | 24 | 10.8 | 7.1-15.6 |
| Parenteral | 198 | 89.2 | 84.5-92.9 |
| If Parenteral, Type | | | |
| Intravenous intermittent | 177 | 89.4 | 84.6-93.2 |
| Intravenous continuous infusion | 20 | 10.1 | 6.3-15.1 |
| Other | 1 | 0.5 | 0.0-2.8 |
| **Parenteral to oral switch** | 204 | | |
| Yes | 6 | 2.9 | 1.3-6.2 |
| No | 195 | 95.6 | 91.9-97.9 |
| Unknown | 3 | 1.5 | 0.3-4.2 |
| **Duration of Antibiotics** (Days) [Mean ±SD] | 3.35 | ±2.8 | |
| **Documented STOP/ REVIEW Order** | | | |
| Yes | 219 | 99.1 | 96.9-99.8 |
| No | 2 | 0.9 | 0.1-3.1 |
| **Compliance with local guidelines** | No local/ Institutional Antimicrobial guidelines available | | |

CAI: Community-Acquired infection MP: Medical Prophylaxis; SP: Surgical Prophylaxis; HAI: Hospital-Acquired Infection.
SP1: Single preoperative dose; SP2: Multiple doses within one day; SP3: Multiple doses beyond 24hours; SPO: Prophylaxis before surgery. IV: Intravenous; SD: Standard Deviation. Parenteral: injectable/infusion administration.

CI: 48.7–62.1%) were the most frequent, followed by medical prophylaxis, MP 63 (28.9%, 95% CI: 23.1–35.3%) and surgical prophylaxis SP 20 (9.2%, 95% CI: 6.0–13.7%). Among 20 patients receiving surgical prophylaxis (SP), 6 (30%) received a single preoperative dose (SP1), 10 (50.0%) received multiple doses within one day (SP2), 1 (5.0%) received multiple doses extending beyond 24 hours (SP3), and 3 (15.0%) received antibiotics for surgical prophylaxis but had not yet undergone surgery (prophylaxis before surgery, SPO). The majority of antimicrobials were administered parenterally

(by injection or infusion) 198 (89.2%, 95% CI: 84.5–92.9%), while 24 (10.8%, 95% CI: 7.1–15.6%) were administered antimicrobials orally. Among parenteral administrations, intravenous intermittent infusion was the most common route 177 (89.4%, 95% CI: 84.6–93.2%), followed by intravenous continuous infusion 20 (10.1%, 95% CI: 6.3–15.1%), and other routes 1 (0.5%, 95% CI: 0.0–2.8%). An intravenous-to-oral switch was documented in only 6 (2.9%, 95% CI: 1.2–6.2%) cases out of 204 eligible ones.

For community and hospital acquired infections (n = 127), empirical therapy (treatment initiated based on clinical judgment before pathogen identification) was used for 126 (99.2%, 95% CI: 95.6–99.9%) cases, while only 1 (0.8%, 95% CI: 0.0–2.6%) received targeted therapy (treatment directed at a specific, identified pathogen). The mean duration of antibiotic therapy was 3.35 ± 2.8 days, and a STOP or review order was documented in 219 (99.1%, 95% CI: 96.9–99.8%) of the prescriptions. The median dosing frequency was two doses per day (range: 1–4). Among all antimicrobial prescriptions, 72 (33.0%) were administered every 12 hours (Q12H), 81 (37.2% every 8 hours (Q8H), 7 (3.2%) every 6 hours (Q6H), 47 (21.6%) every 24 hours (Q24H), 4 (1.8%) every 48 hours (Q48H), and 7 (3.2%) were prescribed as STAT (single, immediate) doses shown in "Table C in S5 Tables".

The prevalence of antimicrobial use in different wards is illustrated in Fig 2. Antimicrobial use was 23.2% in the adult high-risk ward, 20% in adult intensive care, and 16.1% in adult medical ward. Neonatal and pediatric wards accounted for less than 5% of the total antimicrobial use.

## Spectrum of antimicrobials prescribed and WHO AWaRe classification

The detailed distribution of antimicrobial agents prescribed, along with their corresponding ATC codes and WHO AWaRe classification is shown in "Table 3". A total of 222 antimicrobial prescriptions were recorded during the survey. The most

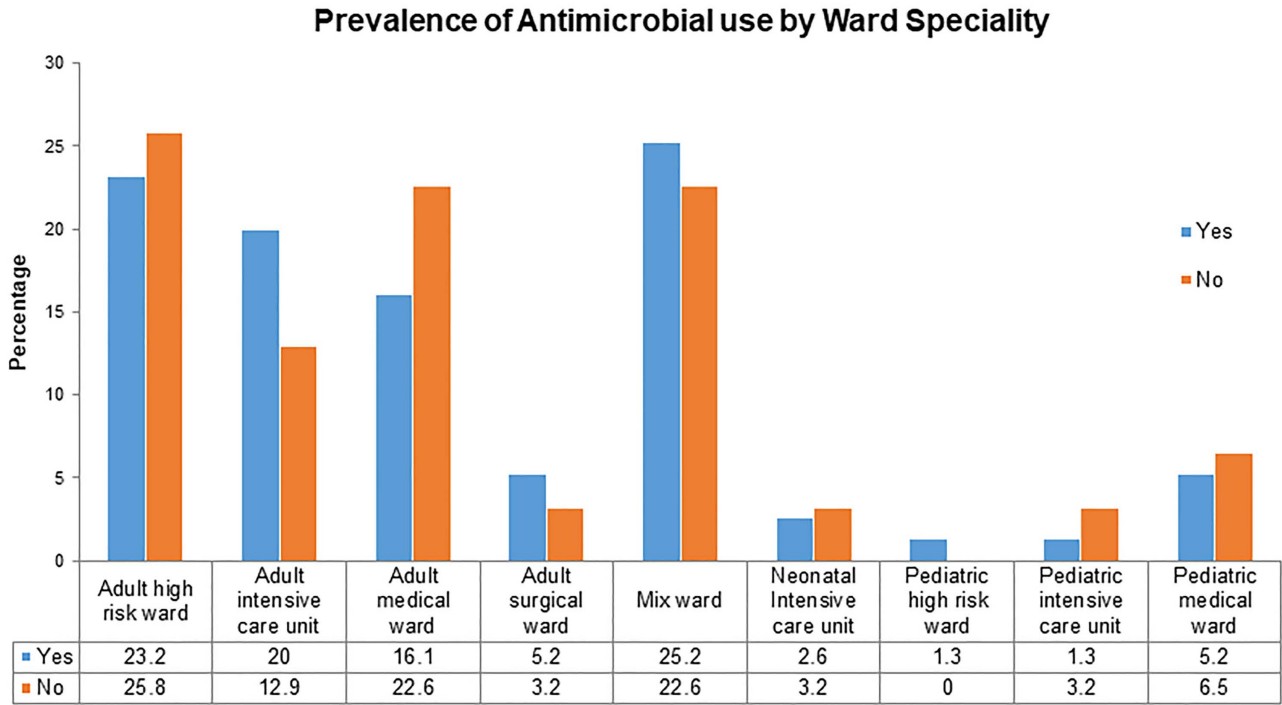

**Prevalence of Antimicrobial use by Ward Speciality**

| | Adult high risk ward | Adult intensive care unit | Adult medical ward | Adult surgical ward | Mix ward | Neonatal Intensive care unit | Pediatric high risk ward | Pediatric intensive care unit | Pediatric medical ward |
|---|---|---|---|---|---|---|---|---|---|
| ■ Yes | 23.2 | 20 | 16.1 | 5.2 | 25.2 | 2.6 | 1.3 | 1.3 | 5.2 |
| ■ No | 25.8 | 12.9 | 22.6 | 3.2 | 22.6 | 3.2 | 0 | 3.2 | 6.5 |

**Fig 2. Distribution of antimicrobial use across inpatient ward specialties.** The graph compares the proportion of cases with antimicrobial treatment (Yes) to those without (No) in various adult, pediatric, mixed, and intensive care wards. Numerical values indicate the percentage for each category.

**Table 3. Spectrum of antimicrobials prescribed and WHO AWaRe classification.**

| Antimicrobials used | ATC Class | WHO AWaRe Class | N = 222 | % | 95% CI |
|---|---|---|---|---|---|
| Amikacin | J01GB06 | Access | 2 | 0.9 | 0.1-3.2 |
| Amoxicillin | J01CA04 | Access | 1 | 0.5 | 0.0-2.0 |
| Amoxicillin/clavulanate potassium | J01CR02 | Access | 8 | 3.6 | 1.8-7.0 |
| Amphotericin | J02AA01 | Not in AWaRe | 2 | 0.9 | 0.1-3.2 |
| Cefazoline | J01DB05 | Access | 1 | 0.5 | 0.0-2.0 |
| Cefoperazone/sulbactam | J01DD62 | Watch | 4 | 1.8 | 0.5-4.0 |
| Cefotaxime | J01DD01 | Watch | 4 | 1.8 | 0.5-4.0 |
| Ceftriaxone | J01DD04 | Watch | 41 | 18.5 | 13.8-24.0 |
| Cefuroxime | J01DC02 | Access | 2 | 0.9 | 0.1-3.2 |
| Ciprofloxacin | J01MA02 | Watch | 7 | 3.2 | 1.4-5.8 |
| Clarithromycin | J01FA09 | Watch | 10 | 4.5 | 2.5-8.5 |
| Clindamycin | J01FF01 | Access | 3 | 1.4 | 0.3-4.3 |
| Co-trimoxazole | J01EE01 | Access | 5 | 2.3 | 1.0-5.2 |
| Doxycycline | J01AA02 | Access | 1 | 0.5 | 0.0-2.0 |
| Gentamicin | J01GB03 | Access | 3 | 1.4 | 0.3-4.3 |
| Imipenem | J01DH51 | Watch | 4 | 1.8 | 0.5-4.0 |
| Levofloxacin | J01MA12 | Watch | 4 | 1.8 | 0.5-4.0 |
| Linezolid | J01XX08 | Reserve | 3 | 1.4 | 0.3-4.3 |
| Meropenem | J01DH02 | Watch | 36 | 16.2 | 11.7-21.6 |
| Metronidazole | J01XD01 | Access | 8 | 3.6 | 3.5-10.1 |
| Moxifloxacin | J01MMA14 | Watch | 1 | 0.5 | 0.0-2.0 |
| Isoniazid/Rifampicin/Ethambutol/Pyrazinamide | J04AM02 (Combo) | Not in AWaRe | 1 | 0.5 | 0.0-2.0 |
| Piperacillin/tazobactam | J01CR05 | Watch | 40 | 18.1 | 13.3-23.7 |
| Rifaxamin | A07AA11 | Access | 6 | 2.7 | 1.2-5.4 |
| Tigecycline | J01AA12 | Reserve | 1 | 0.5 | 0.0-2.0 |
| Vancomycin | J01XA01 | Watch | 24 | 10.8 | 7.3-15.7 |

ATC: Anatomical Therapeutic Chemical classification; AWaRe: WHO classification of antibiotics into Access, Watch, and Reserve group. Access = first-line; Watch = monitored; Reserve = last-resort. Percentages are calculated from total antimicrobial prescriptions (n = 222).

frequently prescribed agents were ceftriaxone 41 (18.5%, 95% CI: 13.8–24.0%) 'Watch', piperacillin/tazobactam 40 (18.1%, 95% CI: 13.3–23.7%) 'Watch', and meropenem 36 (16.2%, 95% CI: 11.7–21.6%) 'Watch'. Other commonly used antimicrobials included vancomycin 24 (10.8%) 'Watch', Clarithromycin 10 (4.5%) 'Watch', and amoxicillin/clavulanate 8 (3.6%) 'Access'.

The overall distribution of antimicrobial agents according to their Anatomical Therapeutic Chemical (ATC) classification and respective frequency of use during the hospital stay are shown in "Table 4". According to the ATC classification, the majority of prescribed agents belonged to the category Antibacterials for Systemic Use (J01), accounting for 205 (92.0%) of all antimicrobials prescribed. Other ATC classes included Antiprotozoals (P01) 8 (3.6%), Antidiarrheals (A07) 6 (2.7%), Antifungals (D01) 2 (0.9%), and Antimycobacterials (J04) 1 (0.45%).

Fig 3 shows the distribution of antibiotics according to the WHO AWaRe classification across different hospital wards. Among adult high-risk wards 24.4% of patients received Watch group antibiotics, while 22.7% of patients in adult intensive care wards and 14% in adult medical wards were also prescribed Watch antibiotics. In Adult ICU 50% of patients were receiving 'Reserve' category antibiotics and 25% in mixed and adult medical wards respectively, whereas in pediatric medical wards and neonatal ICU 2.9% of children were prescribed Access group antibiotics.

**Table 4. Distribution of antimicrobial agents by ATC classification.**

| ATC Classification | Antimicrobial use during hospital stay | Number of Antimicrobial n=222 | % |
|---|---|---|---|
| J01 | Antibacterial for Systemic Use | 205 | 92 |
| D01 | Antifungal | 2 | 0.9 |
| J04 | Antimycobacterials | 1 | 0.45 |
| P01 | Antiprotozoal | 8 | 3.6 |
| A07 | Antidiarrheals | 6 | 2.7 |

ATC: Anatomical Therapeutic Chemical; ATC Classification J01: Antibacterial for Systemic Use; D01: Antifungals; J04: Antimycobacterials; P01: Antiprotozoal; A07: Antidiarrheals.

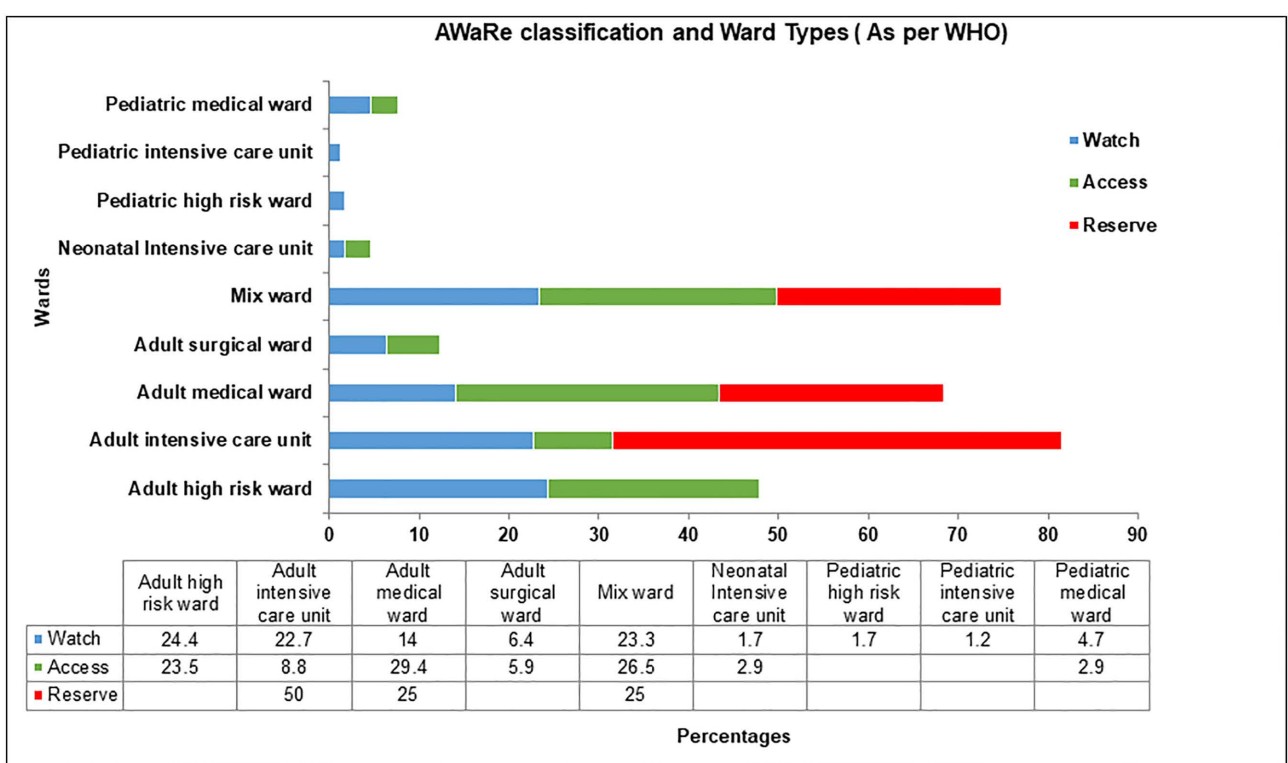

**Fig 3. Distribution of WHO AWaRe categories across hospital ward specialties.** Bars represent the percentage of total antimicrobial prescriptions categorized as **Access** (first-line, narrow-spectrum), **Watch** (higher-risk, broader-spectrum), or **Reserve** (last-resort) antibiotics in each ward. Percentages may not sum to 100% per ward if some prescriptions were unclassified. Data are from a point-prevalence survey.

## WHO AWaRe classification of antibiotics

The distribution of antibiotic prescriptions according to the WHO AWaRe classification is elaborated in Fig 4. Among 219 classified prescriptions, 177 (80.8%, 95% CI: 75.2–85.6%) were Watch group, 38 (17.3%, 95% CI: 12.6–23.2%) were Access group, and 4 (1.8%, 95% CI: 0.7–4.6%) were Reserve group.

## WHO-PPS summary of key findings

"Table 5" shows the summarized key WHO-PPS indicators: overall antibiotic prevalence was 83.3% (155/186 patients, 95% CI: 77.5–88.2%). The most common indication was community-acquired infection 55.5% (121/218, 95%

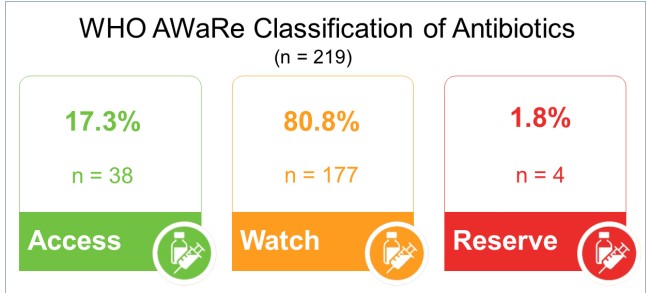

**WHO AWaRe Classification of Antibiotics**
(n = 219)

| Access | Watch | Reserve |
|---|---|---|
| **17.3%** | **80.8%** | **1.8%** |
| n = 38 | n = 177 | n = 4 |

**Fig 4. WHO AWaRe classification of Antimicrobials.** The chart shows the percentage and absolute number (n) of prescribed antibiotics categorized as **Access** (first-line, narrow-spectrum), **Watch** (higher-risk, broader-spectrum), or **Reserve** (last-resort) from a total of 219 prescriptions. Overall distribution of antibiotic prescriptions according to WHO AWaRe classification: Access 17.3% (95% CI: 12.6–23.2%), Watch 80.8% (95% CI: 75.2–85.6%), Reserve 1.8% (95% CI: 0.7–4.6%).

**Table 5. WHO-PPS summary of key antibiotic use indicators.**

| WHO PPS Indicator | % (n/N) | 95% CI |
|---|---|---|
| **Overall antibiotic prevalence** | | |
| Patients receiving ≥ 1 antibiotic | 83.3% (155/186) | 77.5-88.2 |
| **Indications of antibiotic use** | | |
| Community-acquired infection (CAI) | 55.5% (121/218) | 48.7-62.1 |
| Medical prophylaxis (MP) | 28.9% (63/218) | 23.1-35.3 |
| Surgical prophylaxis (SP) | 9.2% (20/218) | 6.0-13.7 |
| Hospital-acquired infection (HAI) | 2.8% (6/218) | 1.3-6.0 |
| **AWaRe category distribution** | | |
| Access | 17.3% (38/219) | 12.6-23.2 |
| Watch | 80.8% (177/219) | 75.2-85.6 |
| Reserve | 1.8% (4/219) | 0.7-4.6 |
| **Surgical prophylaxis duration** | | |
| Single dose (SP1) | 30.0% (6/20) | 13.2-56.0 |
| Multiple dose, ≤ 24h (SP2) | 50.0% (10/20) | 27.2-72.8 |
| Multiple dose, > 24h (SP3) | 5.0% (1/20) | 0.3-24.7 |
| **Route of administration** | | |
| Parenteral (IV/IM) | 89.2% (198/222) | 84.5-92.9 |
| Oral | 10.8% (24/222) | 7.1-15.6 |
| **IV-to-Oral Switch rate** | | |
| Eligible patients switched | 2.9% (6/204) | 1.3-6.2 |
| **Compliance with local guidelines** | | |
| Institutional guidelines available | No (0%) | – |

This table presents core indicators as recommended by the WHO Point Prevalence Survey methodology v1.1. Denominators vary per indicator as per WHO protocol. CAI: Community-acquired infection; HAI: Hospital-acquired infection; MP: Medical prophylaxis; SP: Surgical prophylaxis; AWaRe: WHO Access, Watch, Reserve classification; IV: Intravenous; IM: Intramuscular. SP1, SP2, SP3, SPO categories are defined in the WHO manual.

CI: 48.7–62.1%), followed by medical prophylaxis 28.9% (63/218, 95% CI: 23.1–35.3%) and surgical prophylaxis 9.2% (20/218, 95% CI: 6.0–13.7%). According to the WHO AWaRe classification, 80.8% (177/219, 95% CI: 75.2–85.6%) of antibiotics were Watch category, 17.3% (38/219, 95% CI: 12.6–23.2%) Access, and 1.8% (4/219, 95% CI: 0.7–4.6%) were Reserve. Surgical prophylaxis exceeding 24 hours (SP3) was administered in 5.0% (1/20) of surgical prophylaxis cases. Parenteral administration accounted for 89.2% (95% CI: 84.5–92.9%) (198/222) of antibiotic therapies, with an IV-to-oral switch rate of only 2.9% (95% CI: 1.3–6.2%) (6/204 eligible patients). Compliance with local guidelines could not be assessed as no institutional antimicrobial guidelines were available.

## Microbiological data

The distribution of microbiological specimens collected during the study are summarized in "Table 6". A total of 232 specimens were processed for culture and sensitivity testing. The most frequent specimen types were blood 99 (53.2%, 95% CI: 46.6–59.7%) and urine 74 (39.8%, 95% CI: 33.3–46.7%). Other specimen sources included sputum 13 (7.0%, 95% CI: 3.8–11.6%), respiratory aspirates 13 (7.0%, 95% CI: 3.8–11.6%), fluids 10 (5.4%., 95% CI: 2.6–9.7%), wound/pus 9 (4.8%, 95% CI:2.2–8.8%), cerebrospinal fluid, CSF 7 (3.0%, 95% CI: 1.5–7.6%), tissue 4 (2.2%, 95% CI: 0.6–5.4%), and stool 3 (1.6%, 95% CI: 0.3–4.6%).

Among the 108 culture reports available, no microbial growth was observed in 70 (64.8%, 95% CI: 55.2–73.6%) samples. "Table 7" summarizes the organisms isolated from positive cultures. The identified pathogens were Coagulase-negative staphylococci 11 (10.2%, 95% CI: 5.2–17.6%) and *Enterococcus* spp. 5 (4.6%, 95% CI: 1.5–10.5%), followed by *Escherichia coli* 4 (3.7%, 95% CI: 1.5–9.2%), *Pseudomonas aeruginosa* 4 (3.7%, 95% CI: 1.5–9.2%), *Acinetobacter* spp. 3 (2.8%, 95% CI: 0.6–7.9%), *Proteus vulgaris* 3 (2.8%, 95% CI: 0.6–7.9%), *Staphylococcus aureus* 3 (2.8%, 95% CI: 0.6–7.9%), *Klebsiella pneumoniae* 3 (2.8%, 95% CI: 0.6–7.9%), and single isolates of *Salmonella typhi* 1 (0.9%, 95% CI: 0.0–5.0%) and *Stenotrophomonas maltophilia* 1 (0.9%, 95% CI: 0.0–5.0%).

## Stewardship opportunities and compliance with guidelines

Antimicrobial stewardship interventions were deemed applicable in 123 (55.4%, 95% CI: 48.7–62.0%) of the 222 antimicrobial prescriptions assessed, illustrated in "Table 8". The most common reasons justifying stewardship interventions were over-prescribing 37 (30.1%, 95% CI: 22.4–39.0%), incorrect dosing 19 (15.4%, 95% CI: 10.0–22.9%), and overlapping antimicrobial spectrum 17 (13.8%, 95% CI: 8.8–21.1%). Other contributing factors included narrow-spectrum options available but not used 18 (14.6%, 95% CI: 8.8–21.1%), inappropriate choice of antibiotic 11 (8.9%, 95% CI: 4.6–15.4%), no indication mentioned 22 (17.9%, 95% CI: 12.0–25.7%), unnecessary escalation

Table 6.  Distribution of microbiological specimens.

| Specimen Type | N = 232 | % | 95% CI |
|---|---|---|---|
| Wound/Pus | 9 | 4.8 | 2.2-8.8 |
| Blood | 99 | 53.2 | 46.6-59.7 |
| Urine | 74 | 39.8 | 33.2-46.7 |
| CSF | 7 | 3.0 | 1.5-7.6 |
| Stool | 3 | 1.6 | 0.3-4.6 |
| Tissue | 4 | 2.2 | 0.6-5.4 |
| Fluid | 10 | 5.4 | 2.6-9.7 |
| Sputum | 13 | 7.0 | 3.8-11.6 |
| Respiratory Aspirate specimen | 13 | 7.0 | 3.8-11.6 |

CSF: Cerebrospinal Fluid. Percentages may exceed 100% due to multiple specimens per patient.
95% CIs calculated using the Wilson score method having number of patients as denominator.

**Table 7. Microorganisms isolated from positive cultures.**

| Organism | N = 108 | % | 95% CI |
|---|---|---|---|
| *Acinetobacter* spp. | 3 | 2.8 | 0.6-7.9 |
| Coagulase-negative staphylococci | 11 | 10.2 | 5.2-17.6 |
| *Escherichia Coli* | 4 | 3.7 | 1.5-9.2 |
| *Enterococcus* spp. | 5 | 4.6 | 1.5-10.5 |
| *Pseudomonas aeruginosa* | 4 | 3.7 | 1.5-9.2 |
| *Proteus vulgaris* | 3 | 2.8 | 0.6-7.9 |
| *Staphylococcus aureus* | 3 | 2.8 | 0.6-7.9 |
| *Salmonella typhi* | 1 | 0.9 | 0.0-5.0 |
| *Stenotrophomonas maltophilia* | 1 | 0.9 | 0.0-5.0 |
| *Klebsiella pnuemoniae* | 3 | 2.8 | 0.6-7.9 |
| No Growth | 70 | 64.8 | 55.2-73.6 |

No growth was observed in 64.8% of cultures. Percentages are based on total positive cultures (n = 108). 95% CIs calculated using the Wilson score method.

**Table 8. Antimicrobial stewardship opportunities and reasons (n = 222).**

| Variables | N (Yes) | % | 95% CI |
|---|---|---|---|
| Antimicrobial stewardship applicable? | 123 | 55.4 | 48.7-62.0 |
| Reasons | | | |
| Over-prescribing | 37 | 30.1 | 22.4-39.0 |
| Incorrect dose | 19 | 15.4 | 10.0-22.9 |
| Overlapping spectrum | 17 | 13.8 | 8.8-21.1 |
| Inappropriate choice | 11 | 8.9 | 4.6-15.4 |
| Missed narrow-spectrum opportunity | 18 | 14.6 | 8.8-21.1 |
| Unnecessary escalation/addition of other antibiotic | 9 | 7.3 | 3.5-13.4 |
| No Indication | 22 | 17.9 | 12.0-25.7 |
| Extended surgical prophylaxis | 8 | 6.5 | 2.9-12.3 |
| Unjustified prolonged duration of therapy | 7 | 5.7 | 2.3-11.3 |
| Broad-Spectrum Antibiotic | 6 | 4.9 | 1.8-10.3 |
| Drug Bug Mismatch | 8 | 6.5 | 2.9-12.3 |
| Use of restricted antibiotic without ID approval | 6 | 4.9 | 1.8-10.3 |
| Microorganism resistant to antibiotic used | 1 | 0.8 | 0.0-4.4 |
| Total stewardship opportunities | 160* | 72.1%* | 65.7-77.9 |

Stewardship opportunities were identified based on WHO PPS methodology v1.1. Yes: means antibiotic stewardship application was found in the prescriptions. *Multiple reasons could apply per prescription, so total n > 123 ID: Infectious Disease.

or addition of another antibiotic 9 (7.3%, 95% CI: 3.5–13.5%), extended surgical prophylaxis 8 (6.5%, 95% CI: 2.9–12.3%), drug–bug mismatch 8 (6.5%, 95% CI: 2.9–12.3%), unjustified prolonged duration of therapy 7 (5.7%, 95% CI: 2.3–11.3%), broad-spectrum antibiotic use 6 (4.9%, 95% CI: 1.8–10.3%), restricted antibiotic use without ID approval 6 (4.9%, 95% CI: 1.8–10.3%), microorganism resistant to antibiotic used 1 (0.8%, 95% CI: 0.0–4.4%). It was noted that no local guidelines were available for antimicrobial use in the hospital. Consequently, the hospital lacks standardized protocols for antimicrobial prescribing, and compliance with guidelines could not be assessed during the PPS.

## Discussion

The findings of this study reveal an alarmingly high and irrational use of antibiotics, mainly due to over-reliance on 'Watch' category, high use of empiric therapy and excess parenteral administration of antibiotics. This study adds a contemporary data point to the growing evidence from Pakistan, which includes prior surveys in Karachi and other public tertiary care settings [16,32]. Our results from a major tertiary care hospital in Karachi underscores a persistent and critical gap in anti-microbial stewardship, warranting urgent, targeted interventions to deal with the rising threat of antimicrobial resistance in a major public healthcare setting.

The patient cohort represents snapshot of typical inpatient in a public tertiary care hospital in Karachi. The gender distribution and mean age as shown in "Table 1", are comparable to cohorts of other PPS conducted in Pakistan, suggesting our findings are generalizable within similar settings [16,38]. Most importantly, the surveyed population consisted mostly of direct admission (94.6%) with no recent hospitalization (92.5%), "Table 1", suggesting that the surveyed population was mostly new inpatients, which differs from the findings of other studies in Pakistan that included referred patients [16,32], indicating that the high antibiotic burden we observed was not due to complex, multi-drug resistant infections due to chronic healthcare exposure. On the contrary, it reflects the prescribing patterns for community-onset acquired conditions. The high prevalence of patients with medical devices (peripheral vascular catheter 87.6%, and urinary catheter 50.5%), highlights a significant factor influencing risk of infection and practice commonly observed in hospitals, that may influence clinicians to initiate antibiotic therapy, a factor that must be considered while designing a targeted ASP [39,40].

Given that our patient population was largely comprised of new admissions, the observed point prevalence of antibiotics use, 83.3%, is particularly striking "Table 2". This percentage is notably higher than previously reported 75% in a previous multi-center point prevalence survey in Pakistan [32] and substantially exceeds the global average of 34.4% [41]. While this prevalence rate aligns with other studies conducted in Pakistan, one of them reporting 90% in a large public sector hospital in Lahore, and similar higher rate of 77% and 89% in other hospitals in Punjab [16], the significance lies in what it reveals about local prescribing culture. The contributing factors are likely multifaceted: a high burden of infectious diseases in a tertiary care environment, empirical prescribing of broad spectrum agents as a defensive measure against diagnostic uncertainty, the unstable medical condition of the patient population in our tertiary care setting, the absence of institutional guidelines or policies to curb unnecessary use. This culture of over-prescription is not just a statistical outlier but a primary, modifiable driver of antibiotics selection in our hospital, warranting monitoring and targeted interventions to promote rationale use of antibiotics. Therefore, reducing this prevalence must be the primary goal of ASP, requiring interventions to address both prescribing behaviors and systems enabling overuse.

The widespread antibiotic use was characterized by a heavy reliance on a narrow spectrum of high-potency agents. Ceftriaxone, Piperacillin/tazobactam, Meropenem- all Watch-group agents, accounted for over half of all prescriptions "Table 3", this percentage aligns with finding from other studies in Pakistan and India, where carbapenems and advanced-spectrum β-lactams dominates the hospital formularies because of their higher perceived reliability and ease of use [32,42,43]. The clinical significance of this pattern puts an intense selection pressure on ESBL (Extended-spectrum Beta-lactamase) and carbapenemase-producing organisms within the hospital, and also reduces the therapeutic utility of these critical drugs for future infections. This pattern points to systemic drivers, including a formulary favoring or not restricting broad-spectrum antibiotics, lack of prescriber confidence in narrow-spectrum antibiotic options, and a risk-averse clinical culture of the prescribers.

The cumulative effect of these practice are evident by the distorted WHO AWaRe classification in our setting. An extraordinary 80.8% of all antibiotics prescribed belonged to 'Watch' category, double the WHO recommended target of 40% [44]. While 'Access' category accounted for only 17.3% "Fig 4". This distribution of antibiotics is considerably higher than the figures reported in the other studies, i.e., 72.1% in 'Watch' and 25.7% in 'Access' in pediatric patient in Punjab [45], and 57.03% in the 'Watch' group and 32.67% in 'Access' group in India [43]. Comparison of these results with other

Low-and middle-income countries (LMICs) also highlights this concern, e.g., a study from Democratic Republic of Congo reported 'Watch' group 43.2% while 36.5% in 'Access' group [46]. The high use of 'Watch' group is a concern especially in LMICs, as this group comprises broad-spectrum antibiotics having high potential of resistance [47], and are supposed to be used as second-line options in patients with confirmed cultures [42]. This extreme distortion in our setting points out that the 'Watch' category has effectively became the *de facto* first line choice, a practice that can be seen as a primary driver of AMR and this practice weakens our last-line of defense, i.e., 'Reserve' agents and leading to untreatable infections. Several factors drive the over-reliance on 'Watch' group. First, although 'Access' agents were available in the hospital formulary, they were underutilized compared to broad-spectrum antibiotics, indicating a gap in prescriber awareness of AWaRe, lack of compliance to restriction policies and perceived therapeutic reliability encourages their empirical use. Second, the extensive use of empiric therapy (99.2%) along with high negative culture rates (64.8%) reflects a grey area, where lack of timely microbiology results encourages the use of broad-spectrum antibiotics. Finally, absence of local guidelines and formulary restrictions provide no safety against this pattern. This over reliance on 'Watch' Category can be corrected by developing certain strategies, e.g., implementing local guidelines that promote the 'Access' agents as first line, to conduct prospective audits for the justification of 'Watch' group use and restrict the access to' Watch/Reserve' agents, elaborated in "Table 9".

The extensive use of potent broad-spectrum agents is further entrenched by their intravenous route of administration. An 89.2% rate of parenteral administration "Table 2", combined with only a 2.9% IV-to-oral switch rate "Table 5", indicates that these antibiotics are not only chosen first but are also continued longer than clinically necessary, reflecting a trend observed in LMICs. The drivers of the excess use of intravenously administered drugs include severity of infection, physician preference and patient perceptions [48]. A study from Nepal reported a rate 88.5% administration of parenteral antibiotics [49], another study from Malaysia also reported high rate of parenteral antibiotics 62.7%, even when not supported by evidence [48]. Although patients with severe infection need parenteral antibiotics, high rates as observed in this study, often indicate missed opportunities to IV-to-oral switch. Parenteral-to-oral switch of antimicrobials is recognized as a key parameter of antibiotic stewardship process, reducing costs and catheter related complications [48]. The missed opportunities for IV-to-oral switch have direct negative consequences: they prolong hospital stays, increase the risk of catheter associated blood infections, and significantly raise healthcare costs without improving patient outcomes. Interventions like reinforcement of the existing institutional IV-to-Oral switch guidelines, introduction or automatic alerts in patient records after 48–72 hours of IV therapy to review for oral switch, and education of prescribers on benefits and pharmacokinetic of oral alternatives can help curb this issue as summarized in "Table 9".

Our study revealed an exceptionally high reliance on empirical treatment (99.2%), paired with 64.8% culture-negative rate, pointing out a critical failure in diagnostic stewardship, the high culture negative rate is consistent with the reports from Pakistan and other LMICs showing 40–90% culture negative across urine, CSF, and surgical specimens [50–53]. This high rate of no-growth cultures is likely influenced by empirical antibiotic use prior to sample collection, suboptimal timing of specimen collection, limitations of routine laboratory methods, and gaps in diagnostic stewardship [54,55]. Our findings are also consistent with a WHO point prevalence survey at a tertiary care center in Pakistan, where only 23.9% of specimens were culture-positive and nearly 70% were culture-negative, highlighting the limited yield of routine microbiological diagnostics in similar hospital settings [16]. This over-reliance on empirical treatment, with low percentages of antibiotic prescription based on culture and sensitivity report in the region and other LMICs reflects a general trend in the region [45,56,57]. We acknowledge that the exact timing of microbiological sample collection in relation to antibiotic administration was not systematically documented in patient records, which may have influenced culture positivity rates and antimicrobial susceptibility results. Prolonged microbiology turnaround time, specimens collected after start of antibiotic therapy, and limited access to proper microbiological facilities for identifying the pathogen and their susceptibility encourage clinicians to default to broad-spectrum antibiotics rather than waiting for the culture guided use of antibiotics. This practice, observed in

 

**Table 9. Summary of key irrational prescribing practices identified and proposed stewardship intervention.**

| Key finding from PPS | Clinical and AMR implication | Proposed Stewardship intervention |
|---|---|---|
| Absence of local antimicrobial guidelines | Leads to non-standardized, variable, and often irrational prescribing | 1. Reinforce the role and functioning of existing AMS committee to develop evidence-based local treatment guidelines |
| | | 2. Disseminate guidelines via hospital intranet, pocket cards or handbooks, and educational sessions |
| High use of Watch Category antibiotics (80.8%) | Drives antimicrobial resistance; compromises efficacy of last-resort drugs | 1. Continuous promotion of Access group agents as first line |
| | | 2. Introduce prospective audit and feedback with mandatory justification for watch group use |
| | | 3. Reinforce the policy of restrict access to certain Watch/Reserve agents (e.g., require ID approval) |
| Overwhelming empirical therapy (99.2%) with low culture yield (64.8% no growth) | Inappropriate spectrum coverage; increased risk of toxicity and resistance | 1. Enhance Diagnostic stewardship: Improve pre-analytical phases (specimen collection training, timing) |
| | | 2. Invest in rapid diagnostic tests (e.g., multiplex PCR) |
| | | 3. Implement a 48–72 hours "auto stop"for empirical therapy, mandating review and de-escalation |
| Missed opportunities for IV-to-Oral switch (only 2.9%) | Prolongs hospital stay, increases risk of catheter-related bloodstream infections, and raises costs | 1. Promote the implementation of institutional IV-to-Oral switch guidelines |
| | | 2. Introduce automatic alerts in patient records after 48–72 hours of IV therapy to review for switch eligibility |
| | | 3. Educate prescribers on the benefits and pharmacokinetics of oral alternatives |
| High prevalence of antimicrobial use (83.3%) | Indicates a culture of over-reliance on antibiotics, contributing to overall AMR burden | 1. Conduct regular PPS to monitor trends and provide feedback to departments |
| | | 2. Launch institution-wide awareness campaign on AMR |
| | | 3. Continuous monitoring of the institutional quality indicators of antimicrobial prescribing |

AMR: Antimicrobial Resistance; AMS: Antimicrobial Stewardship; ID: Infectious Diseases; PCR: Polymerase Chain Reaction; PPS: Point prevalence Survey. Interventions are proposed based on study findings and WHO recommendations.

India and other LMICs, undermines stewardship efforts and accelerates antimicrobial resistance [58]. This high rate of empiric therapy is not just a prescribing pattern but the root cause resulting in the overuse of 'Watch' group antibiotics and prolonged IV therapy documented earlier. Improving diagnostics by strategies like improved sample collection protocols, strengthened laboratory capacity, and robust ASP to optimize diagnostic yield is equally important as antibiotic policies and guidelines [59]. On the other hand implementation of auto-stop orders for empiric therapy at 48–72 hours can mandate physicians to review and de-escalate the therapy. Empirical therapy is necessary, but its use should be informed and adjusted by timely diagnostics.

The indication for antibiotic use in this study shows a pattern where prophylaxis contributes significantly to the overall burden of antibiotic use. CAIs were the leading documented reason for antibiotic use (55.5%), followed by medical prophylaxis (28.9%) as shown in "Table 2". The proportion of medical prophylaxis reported is higher than the rates reported in the LMICs and other PPS in Pakistan [43,45], suggesting that antibiotics are used preventively in clinical setting not strongly supported by evidence, such as in patients with indwelling catheters without clear sign of infection [43]. While surgical prophylaxis accounted for a lower proportion (9.2%) "Table 2", compared to India and other Pakistani studies, the extended duration beyond 24 hours in some cases indicates a common misunderstanding of the protocols [32,43]. This pattern highlights that antimicrobial overuse is not only related to treatment of active infection but also for preventive practices. The high use of prophylactic antibiotics without evidence-based guidelines, can contribute to AMR, establishing a clear need of evidence based antibiotic guidelines and adherence to rationale use of antibiotics and restrict non-therapeutic antibiotic use.

These widespread prescribing irregularities are not surprising given that there are no institutional guidelines for the use of antibiotics in the hospital. The finding that 55.4% of prescriptions had a stewardship opportunity-with reasons being over-prescribing, incorrect dose, and overlapping spectrum, is a direct measure of dysfunction and widespread irrational prescribing practices. This is not a failure of individual prescribers, but of the hospital system, which lacks standardized protocols for antibiotic use. This finding is not unique for Pakistan where studies highlight absence of guideline or policy for 61.9% antimicrobial agents, and lack of local protocols and guidelines [32]. This structural gap explains why problematic practices exist and underscores that the foremost stewardship intervention may be development and implementation of evidence-based local guidelines, tailored to the hospital's specific resistance patterns and cases. Other strategies like regular monitoring of antibiotic use through periodic PPS to monitor the trends, providing feedback to physicians, including antimicrobial prescription as the quality indicators of the hospital and strengthening of institution wide awareness campaign on AMR can help reduce the number of prescriptions requiring stewardship "Table 9".

The findings of this study strongly advocate for the immediate establishment and strengthening of ASP in the tertiary care hospitals of Karachi. The high prevalence of antibiotic use, the dominance of 'Watch'group, the extensive empirical therapy, and the practice of irrational prescribing, present targets for stewardship interventions. Other studies in Pakistan have also emphasized the rudimentary nature of ASPs and the need for structured programs to improve antibiotic utilization [16,45,60]. Implementation of ASPs in LMICs faces significant challenges, including shortage of human resource, diagnostic deficiencies and poor monitoring capacities [45,52,61,62]. Successful implementation of ASPs in other LMIC settings, such as in Vietnam, has demonstrated significant reductions in antibiotic consumption and costs, providing a model for potential adoption [63]. Despite these barriers, the implementation of WHO's PPS methodology, as demonstrated in this study, is a crucial first step to identify these gaps and inform evidence-based strategies [32]. Effective ASPs are seen as vital to optimizing antibiotic prescribing and reducing AMR.

Our findings advocate the immediate strengthening of a multifaceted ASP tailored to the local context. Key interventions should include:

- Developing and mandating adherence to local evidence-based antimicrobial guidelines based on hospital data

- Implementing prospective audit and feedback with intervention, focusing on de-escalation from 'Watch' to 'Access' agents and promoting IV-to-oral switch.

- Enhancing diagnostic stewardship to improve culture yield and turnaround times.

- Enhancing the role of dedicated multidisciplinary ASP team to lead these efforts along with continuous prescriber education.

This study has several strengths, including the use of WHO PPS methodology, this standardized approach ensures the comparability of findings with other national and international studies [48,64]. We used patient medical records of the hospital that reflects actual clinical practices, and provide a true snapshot [65]. The data collection was carried out by an ID physician and a pharmacist, suggesting a collaborative effort in data acquisition. The study successfully identified specific areas for intervention like, 55.4% cases as "Antimicrobial stewardship applicable", "no local guidelines", prescribing patterns, making the outcomes of the study as a building block to strengthen the ASP in the hospital. As per the design, i.e., PPS, the study focuses on prescribing patterns, and does not collect or report any patient-specific outcomes like treatment success, adverse drug reactions, or development of resistance, making it a limitation to report patient-specific outcomes [48,64], future longitudinal studies are recommended to track trends in prescribing, and interventional studies are required to assess the impact of the ASP strategies recommended in this study. The study relies on the "data from patient medical record", this could also be a limitation in the complete analysis due to missing data [45]. Specifically, the timing of microbiological specimen collection relative to antibiotic initiation was not consistently documented, limiting our ability to fully assess its impact on culture yield. The unavailability of local antimicrobial guidelines limits this study to identify deviations

from desired local practices. The study no doubt uses the standardized protocol, using an ID physician and a pharmacist can introduce a degree of interpretation and potential observer bias, particularly dealing with incomplete medical records [48]. Finally, as this study was conducted in one of the tertiary care public hospitals in Karachi, so the findings may not be fully generalizable to other hospitals in the region, particularly with established ASPs or different population.

## Conclusion

The study reveals overreliance on antibiotics in tertiary care hospitals in Karachi, especially excessive use of 'Watch' category agents, high empirical use of antibiotics, and missed IV-to-oral switch. These irrational prescribing practices in the absence of local guidelines highlight a critical need for implementing and strengthening ASP. The findings serve as an urgent call to action for immediate strengthening of ASP, focused on development of local guidelines, enhancing diagnostic stewardship, and rationalizing antibiotics selection and administration to combat the escalating threat of AMR and improve patient outcomes.

## Supporting information

**S1 Appendix. Patient case report tool.** Form used for collecting patient demographic and clinical data.
(DOCX)

**S2 Appendix. Antibiotic use form.** Form documenting antibiotic prescriptions, including drug name, dose, route, duration and indication.
(DOCX)

**S3 Appendix. Microbiology data form.** Form for recording microbiological specimen details, culture results, and susceptibility data.
(DOCX)

**S4 Appendix. Stewardship assessment form.** Checklist used to identify and categorize antimicrobial stewardship opportunities based on predefined criteria including incorrect dosing, spectrum overlap, unnecessary prolonged prophylaxis, absence of indication, inappropriate antibiotic selection and missed IV-to-oral switch.
(DOCX)

**S5 Tables. Supplementary tables.** Table A – showing demographic and clinical details. Table B – detailing descriptive data of the wards. Table C – providing description of route of administration, frequency and site of antibiotics administration.
(DOCX)

**S1 Fig. Graphical abstract: Point prevalence survey.** Snapshot of antibiotic use patterns and stewardship priorities from a hospital point prevalence survey.
(TIF)

## Author contributions

**Conceptualization:** Ale Zehra, Tehreem Ansari.

**Data curation:** Ale Zehra, Tehreem Ansari, Syed Shaukat Ali Muttaqi Shah.

**Formal analysis:** Syed Shaukat Ali Muttaqi Shah, Beenish Syed, Farah Saeed.

**Funding acquisition:** Ale Zehra, Tehreem Ansari, Mehwish Rizvi, Fakhsheena Anjum.

**Investigation:** Ale Zehra.

**Methodology:** Ale Zehra, Tehreem Ansari, Tehrim Fatima.

**Project administration:** Ale Zehra, Tehreem Ansari, Mehwish Rizvi, Fakhsheena Anjum.

**Resources:** Tehreem Ansari, Beenish Syed.

**Software:** Syed Shaukat Ali Muttaqi Shah, Mehwish Rizvi, Fakhsheena Anjum.

**Supervision:** Ale Zehra, Tehreem Ansari.

**Validation:** Ale Zehra, Syed Shaukat Ali Muttaqi Shah, Mehwish Rizvi, Fakhsheena Anjum.

**Visualization:** Syed Shaukat Ali Muttaqi Shah, Beenish Syed, Mehwish Rizvi, Fakhsheena Anjum, Farah Saeed.

**Writing – original draft:** Ale Zehra, Tehreem Ansari, Mehwish Rizvi, Tehrim Fatima.

**Writing – review & editing:** Ale Zehra, Tehreem Ansari, Syed Shaukat Ali Muttaqi Shah, Beenish Syed, Tehrim Fatima, Farah Saeed.

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
