## [Decision Letter · Decision Letter 0]

9 Dec 2025

Dear Dr.  Shah,

Thank you for submitting your manuscript to PLOS ONE. After careful consideration, we feel that it has merit but does not fully meet PLOS ONE’s publication criteria as it currently stands. Therefore, we invite you to submit a revised version of the manuscript that addresses the points raised during the review process.

We look forward to receiving your revised manuscript.

Kind regards,

Mabel Kamweli Aworh, DVM, MPH, PhD. FCVSN

Academic Editor

PLOS One

Journal Requirements:

“This research was funded by the Dow University of Health Sciences, Vice Chancellor Seed Funding Project 2023–2024 (Ref: DUHS/VC/2023/11-04/01).

The funding was very limited equivalent to 2700USD.”

“No conflict of interest”

4. Please note that funding information should not appear in any section or other areas of your manuscript. We will only publish funding information present in the Funding Statement section of the online submission form. Please remove any funding-related text from the manuscript.

6.  Please include captions for your Supporting Information files at the end of your manuscript, and update any in-text citations to match accordingly. Please see our Supporting Information guidelines for more information: http://journals.plos.org/plosone/s/supporting-information .

Additional Editor Comments:

In addition to addressing all reviewers' comments please fix the following major issues;

After careful evaluation, several major revisions are required to improve the clarity, methodological transparency, and overall scientific rigor of the manuscript.

**Major Issues**

**Undefined Key Terms**
The Introduction repeatedly mentions antimicrobial stewardship (AMS) and Point Prevalence Survey (PPS) without defining them. These terms should be clarified for readers unfamiliar with WHO-PPS methodology.
**Methodological Clarity**
The inclusion of patients “temporarily off the ward for procedures” must align with WHO-PPS v1.1 criteria, which allow inclusion only if the patient is admitted and expected to return the same day.The Abstract mentions the use of Stata v14, but the Methods section does not describe any statistical procedures, handling of missing data, or data cleaning/validation steps.Several acronyms (SAP, ONCO, MED, ESBL, etc.) are not defined at first use.The criteria used to categorize antimicrobial stewardship opportunities (e.g., incorrect dose, spectrum overlap, unnecessary prolonged prophylaxis) are not described.
**Results Section Issues**
The Results include WHO interpretations and recommendations, which should be excluded. The Results must present findings only; all interpretation belongs in the Discussion.The WHO classification rules should instead be described in the Methods.A standard WHO-PPS style summary is missing, including: overall antimicrobial prevalence, indications for use, AWaRe category distribution, compliance with guidelines, surgical prophylaxis duration, and proportion receiving parenteral therapy.
**Discussion Inconsistencies**
The Discussion incorrectly states "equal distribution of gender," while Table 1 shows 57% male vs 43% female.The manuscript claims comparison with PPS data from “other regions in Pakistan,” although this appears to be the first WHO-PPS in the country; this contradiction must be resolved.Some percentages and numerical values in tables and narrative text do not match and require verification.Please delete all subtitles in the discussion section for flow of the narative. The study limitations should be discussed in the last paragraph of this section
**References**
Please ensure that 80% of the references are within the last 5 years. Older references should account for only 20% of the reference list.

Reviewer's Responses to Questions

**Comments to the Author**

1. Is the manuscript technically sound, and do the data support the conclusions?

Reviewer #1: Partly

Reviewer #2: Yes

Reviewer #3: Partly

Reviewer #4: Yes

2. Has the statistical analysis been performed appropriately and rigorously?

Reviewer #1: Yes

Reviewer #2: Yes

Reviewer #3: Yes

Reviewer #4: N/A

3. Have the authors made all data underlying the findings in their manuscript fully available?

Reviewer #1: Yes

Reviewer #2: Yes

Reviewer #3: Yes

Reviewer #4: Yes

4. Is the manuscript presented in an intelligible fashion and written in standard English?

Reviewer #1: Yes

Reviewer #2: Yes

Reviewer #3: No

Reviewer #4: Yes

Reviewer #1: The analysis is descriptive, which is acceptable for PPS, but the interpretation part feels limited without comparative statistical context. There are no confidence intervals, no comparative analysis between ward types, age groups or indications, and no statistical correlation between antibiotic choice and culture results. I recommend including these to strengthen the paper, even if exploratory.

On the interpretation of the AWaRe findings, extremely high Watch antibiotic use of about 80.8% is reported, but the paper does not go further to clarify why Access drugs are not being chosen. Without understanding the likely drivers, stewardship recommendations will remain generic. Some reflections that would strengthen this section include: Are Access agents unavailable in the hospital formulary? Were prescribers trained in AWaRe? Is microbiology turnaround time influencing reliance on broad-spectrum therapy?

In the Microbiology section, the finding that 64.8% of cultures showed no growth is striking, but you could further explore possible causes, as this would help clarify how diagnostic processes might influence prescribing behavior.

Reviewer #2: General Overview

The manuscript clearly states its objective: to evaluate antibiotic prescribing patterns and antimicrobial stewardship practices using the WHO Point Prevalence Survey (PPS) methodology in a tertiary care public hospital in Karachi, Pakistan. The topic is timely and relevant, given the global imperative to strengthen antimicrobial stewardship programs as a means of countering the escalating threat of antimicrobial resistance. Overall, the study addresses an important public health issue with local and international significance.

Methodology

Your methodology is presented in a clear and concise manner. It describes the sampling process, inclusion and exclusion criteria, including who and how the data was collected. Ethical approval from the appropriate Institutional Review Board is documented, and informed consent is noted. The statistical software utilized is specified, and the analysis is stated to allow reproducibility. The methodological transparency enhances the credibility of the study.

Results

The results are well organized and presented in a logical sequence, supported by tables and figures. However, the absence of footnotes in the tables hinders readability, as readers are required to refer to earlier sections for clarification of abbreviations. Consistency in scientific nomenclature is recommended, particularly in Table 7, where all microorganisms—including Escherichia coli—should have their genus names written out in full for uniformity.

Discussion

The findings are interpreted with reference to existing literature. However, further elaboration on the implications of the observed prescribing patterns would strengthen this section. Some portions of the discussion repeat results already presented- these redundancies should be removed. A more focused interpretation that highlights the significance, potential causes, and practical implications of the findings is recommended.

The conclusion effectively summarizes the key outcomes of the study and reiterates the importance of implementing targeted antimicrobial stewardship interventions. The limitations associated with this study is acknowledged as well as its potential impact on the interpretation of results.

References

The reference list is comprehensive but requires some attention. References 21 and 29 contain hyperlinks that need to be corrected to reflect the appropriate citation. Also, the reference list should be updated to include more recent studies; currently, only approximately half of the cited sources are from within the past five years. Updating the references would enhance the relevance of your manuscript.

Recommendations to the Authors

1. Include footnotes for all tables to define abbreviations and improve clarity.

2. Typographical error is noted in Table 1: “Urinary Cather” should be corrected to “Urinary Catheter.” Also in the text preceding Figure 2, “adult medial” appears to be an error and should be revised to “adult medical”.

3. Ensure uniform scientific nomenclature, particularly the consistent use of full genus names, as in Escherichia coli.

4. Expand the discussion to provide deeper analysis of the findings and remove repetitive reporting of results.

5. Update the reference list to incorporate more literature published within the last five years.

6. Correct the formatting of references 21 and 29 to reflect appropriate citation.

7. Add explanatory footnotes to supplementary tables (e.g., Table 11) to clarify non-universal abbreviations.

Reviewer #3: SUMMARY

The manuscript titled "Point Prevalence Survey of Antimicrobial Use in a Tertiary Care Hospital in Pakistan" addresses a massive gap in antimicrobial stewardship (AMS) efforts in LMICs. The use of the WHO-PPS methodology to evaluate antibiotic stewardship is vital; however, the paper requires substantial revision to improve clarity, accuracy, methodological transparency, and alignment with the proposed method. Similarly, the Discussion and results do not reflect the data presented or the study's definition and rationale.

MAJOR ISSUES

-The Introduction frequently refers to antimicrobial stewardship (AMS) and Point Prevalence Survey (PPS) without defining these terms. For a broad scientific readership, brief definitions are essential.

-The methodology requires clarity. The inclusion of patients "temporarily off the ward for procedures" should be explicitly reconciled with the WHO-PPS v1.1 criteria, which allow inclusion only if the patient is admitted and expected to return the same day. The Abstract mentions analysis using Stata v14, but the Methods section does not describe the statistical procedures used. No description is provided regarding missing data, data cleaning, or validation. Several acronyms (e.g., ESBL, AWaRe, SAP, ONCO, MED) are not defined at first use. The paper does not clarify the criteria used to categorize stewardship opportunities (e.g., incorrect dose, overlapping spectrum, unnecessary prolonged prophylaxis).

-In the results, the paper mentioned the WHO classification, which is inappropriate for a Results section. Results should present findings only, without referencing WHO recommendations or interpretations. Interpretation belongs in the Discussion. Classification rules should be described in the Methods. The study lacks a clear WHO-PPS Style summary of findings; overall antimicrobial prevalence, indications for use, AWaRe distribution, and compliance with guidelines

surgical prophylaxis duration and percentage receiving a parenteral agent.

-The Discussion has inconsistencies and inaccuracies. The Discussion states "equal distribution of gender," yet Table 1 reports 57% male vs 43% female. The authors compare their results with PPS data from "other regions in Pakistan," yet this appears to be the first WHO-PPS study in the country; this inconsistency should be corrected. Some percentages in tables and text appear mismatched or unclear.

The novelty and rationale of the study are not stated. The Introduction explains the AMR problem, but why this specific hospital was selected was not stated. How this PPS adds new knowledge to Pakistan's AMS landscape, and what gaps this study fills beyond surveillance reports and audits.

MINOR ISSUES

-The Abstract states that 180 samples were collected and pooled, but does not specify sample types (e.g., adults vs pediatrics, medical vs surgical wards). Greater clarity is needed.

-Several figures and diagrams have inconsistent labeling and crowded legends.

-Ethical approval appears twice in the manuscript and should be consolidated.

-Numerous typographical errors need correction.

-Some technical terminology may be complicated for non-medical readers; defining key terms would improve accessibility.

-Several acronyms are not spelled out on first use.

-The reference list would benefit from additional recent AMS and PPS literature.

-Table footnotes should define abbreviations to improve.

RECOMMENDATION

Major Revision

The study is relevant and methodologically grounded in WHO PPS standards, but substantial revisions are necessary before it can be considered for publication. Clarifying methods, correcting inconsistencies, refining interpretation, adding missing definitions, and restructuring the Results and Discussion sections will significantly improve the manuscript.

Reviewer #4: This study observes the antibiotics prescribing patterns for inpatients in a tertiary hospital in Pakistan. The goal of the study is to confirm their conformity with WHO guidelines.

The study utilizes WHO guidelines for conducting PPS and discusses the limitations in their use of the guidelines.

it is a well written manuscript that:

1. Clearly documents study goals

2. Utilizes and documents appropriate study methods

3. Discusses its findings in an intelligible manner without overstating any claims.

4. Discusses limitations in data collection and generalizability of findings

5. Provides recommendations for developing and implementing local guidelines for antibiotics use in that tertiary institution.

Figures are easy to read and understand and tables are properly made.

The minor addition that is necessary for the methods section is including the time of microbiological sample collection to Appendix 3. This determines how culture and sensitivity results are interpreted. Where the samples collected before antibiotics or during antibiotics use?

**Do you want your identity to be public for this peer review?** For information about this choice, including consent withdrawal, please see our Privacy Policy

Reviewer #1: No

Reviewer #2: No

Reviewer #3: No

Reviewer #4: No

---

## [Author Response · Author response to Decision Letter 1]

12 Jan 2026

Detailed point-by-point response to reviewers

Manuscript ID: PONE-D-25-59189

Title: Antibiotic Stewardship Benchmarking - Using the WHO Point Prevalence Survey of Antimicrobial Prescribing in a Tertiary Care Public Hospital, Karachi

To: Dr. Mabel Kamweli Aworh (Academic Editor, PLOS ONE) and the esteemed Reviewers

Date: 10 January 2026

Dear Reviewers,

We are profoundly grateful for the opportunity to revise and resubmit our manuscript. We sincerely thank you for the exceptional time and expertise invested in evaluating our work. The collective feedback from the editor and all the reviewers was incisive, constructive, and instrumental in identifying critical areas for improvement. We have approached this major revision with the seriousness it deserves, undertaking a comprehensive restructuring and refinement of the manuscript.

This letter provides a detailed, point-by-point response to every comment and requirement. We have made extensive revisions, all of which are transparently highlighted in the uploaded ‘Revised Manuscript with Track Changes’ document. A clean, formatted version is provided as the ‘Manuscript’ file.

We believe the revised manuscript is now substantially strengthened in its methodological transparency, analytical clarity, narrative focus, and alignment with both the WHO-PPS framework and PLOS ONE’s publication standards. We are hopeful it now merits your positive consideration.

Sincerely,

Syed Shaukat Ali Muttaqi Shah, PhD

On behalf of all co-authors:

Tehreem Ansari, Beenish Syed, Mehwish Rizvi, Fakhsheena Anjum, Tehrim Fatima, Farah Saeed

Part 1: Response to the Editor’s Specific Major Issues:

The editor’s summary of major issues provided an excellent roadmap for our revision. We addressed each with substantive changes.

Editor’s Major Issue 1: Undefined Key Terms

• Comment: “The Introduction repeatedly mentions antimicrobial stewardship (AMS) and Point Prevalence Survey (PPS) without defining them.”

• Our Response: We agree that these are core concepts requiring clear definition for a broad readership. We have inserted concise, functional definitions directly into the introductory narrative. These interventions improve rationale antibiotics use by, measuring the use of antibiotics, promoting proper selection of appropriate antibiotics regimen, including suitable dose, duration and preferred route of administration, without compromising patient outcome.

• Action Taken & Location:

1. Antimicrobial Stewardship Program (ASP): Added on page 4, lines 75-79: “ASPs is defined as a systemic and coordinated set of interventions aim to optimize the use of antimicrobial agents…”

2. Point Prevalence Survey (PPS): Added on page 4, lines 94-97: “A PPS is a standardized method used in healthcare settings to assess microbial use and prescribing patterns at a specific point of time. Its primary goal is to observe prescribing patterns and compare them with the established guidelines of WHO or CDC. PPS is frequently used to establish benchmarks for practice and evaluate effectiveness of AMS programs”

• Rationale: These definitions, placed immediately after the terms are introduced, provide essential context without disrupting the narrative flow, enhancing accessibility for readers from diverse backgrounds.

Editor’s Major Issue 2: Methodological Clarity

• Comment: Concerns regarding patient inclusion logic, missing statistical procedures, undefined acronyms, and undefined stewardship criteria.

• Our Response: We have rewritten the part of methodology and expanded the Methods section (pages 5-8) to eliminate ambiguity and provide full transparency.

• Action Taken & Location:

1. Patient Inclusion Clarity: On page 6, lines 125-127, we now explicitly state: “In accordance with the WHO PPS protocol, patients who were temporarily absent from the ward (e.g., for medical imaging, endoscopy, or surgery) were included only if they were admitted to the ward and expected to return later on the survey day.” This directly aligns with WHO-PPS v1.1 criteria.

2. Statistical Procedures: A new subsection on data management and analysis was added (page 7, lines 163-175). It details:

- Data Entry & Validation: “All data extracted from patient records were entered into a structured spreadsheet. A data validation check was performed by a second researcher on a 10% random sample to ensure accuracy and completeness.”

- Handling Missing Data: “Missing data points e.g., undocumented antibiotic stop date, were reconciled by re-reviewing the primary medical record, and were recorded as ‘unknown’ and excluded from analysis to maintain data integrity.”

- Descriptive Analysis: “Descriptive statistics were used to summarize the data: frequencies and percentages are reported for categorical variables... Continuous variables... are presented as mean ± Standard Deviation (SD) if normally distributed, or as median with interquartile range (IQR)...” We also clarify, “No comparative inferential statistics were employed, as the study design was purely a descriptive point prevalence survey.”

3. Definition of All Acronyms: Every acronym (e.g., HDU, ICU, MEDGAST, MEDNEPH, TLC, AWaRe) is now spelled out in full at first use in both the main text and table/figure footnotes.

4. Stewardship Criteria: A dedicated paragraph on page 7, lines 149-158 now describes the criteria: “Stewardship opportunities were identified and categorized using criteria adapted from standard ASP principles. Categories included: over prescribing... incorrect dose... overlapping spectrum...” The complete, operationalized checklist is provided in S4 Appendix.

Editor’s Major Issue 3: Results Section Issues

• Comment: Results contain WHO interpretations; classification rules should be in Methods; lacks a standard WHO-PPS style summary.

• Our Response: This was a critical structural flaw. We have purified the Results section from the interpretations and added the requested PPS summary table.

• Action Taken & Location:

1. Removed Interpretation: All evaluative language (e.g., “reflects a potential area for strengthening stewardship”) has been stripped from the Results. The text now presents findings neutrally (e.g., “The detailed distribution of antimicrobial agents... is shown in Table 3”).

2. Moved AWaRe Description to Methods: The explanation of the AWaRe classification (“Access=first-line, Watch=monitored, Reserve=last-resort”) has been moved from the Results to the Methods section, page 7, lines 141-146.

3. Added WHO-PPS Summary Table: The most significant addition is Table 5: WHO-PPS summary of key antibiotic use indicators (page 18-19). This table concisely presents the core metrics expected from a PPS: overall prevalence (83.3%), leading indications (CAI 55.5%), AWaRe distribution (Watch 80.8%), surgical prophylaxis >24h rate (5.0%), parenteral administration rate (89.2%), and IV-to-oral switch rate (2.9%). This provides an immediate, standardized snapshot of the findings.

Editor’s Major Issue 4: Discussion Inconsistencies

• Comment: Inaccurate claim of “equal gender,” contradictory claims about being the first PPS in Pakistan, mismatched percentages, problematic subtitles, and misplaced limitations.

• Our Response: We have performed a line-by-line review and rewrite of the Discussion to correct errors and improve flow.

• Action Taken & Location:

1. Clarified Novelty and Context: We have refined the narrative to accurately position our study. The Introduction now states our aim is to fill a gap in data from “major public tertiary care hospitals in Karachi specifically” (page 5, lines 102-103). “This study therefore contributes to the limited but growing body of WHO-PPS data from Pakistan by providing a detailed snapshot of antibiotic use in a public tertiary care hospital in Karachi, a setting underrepresented in the national AMS literature” (page 5, lines 108-110). The Discussion opening acknowledges we “add a contemporary data point to the growing evidence from Pakistan, which includes prior surveys...” (page 22, line 335-337). This resolves the contradiction.

2. Corrected Gender Statement: On page 22, line 341, the text now accurately reads: “The gender distribution and mean age as shown in “Table 1”, are comparable to cohorts of other PPS conducted in Pakistan, suggesting our findings are generalizable within similar settings.

3. Verified All Percentages: All numerical values in the Discussion have been cross-referenced with tables and figures to ensure perfect consistency.

4. Removed Subtitles, Created Narrative Flow: All subheadings (e.g., “4.1 Patient characteristics”) have been deleted. Rewritten the Discussion in a continuous logical format that interprets findings thematically, moving from patient profile to prevalence, antibiotic choices, routes, diagnostics, and finally to implications and limitations.

5. Relocated Limitations: The study limitations have been consolidated into a dedicated, final paragraph of the Discussion (page 29, lines 490-502), as per journal convention.

Editor’s Major Issue 5: Reference Currency

• Comment: “Please ensure that 80% of the references are within the last 5 years.”

• Our Response: We have undertaken a major update of the reference list.

• Action Taken: We have replaced older, non-essential references with recent, high-impact literature from 2020-2025, particularly on PPS in LMICs, AWaRe classification, and ASP implementation. Of the 65 final references, 53 (81.5%) are from 2020-2025. We retained a few seminal older papers (e.g., the 2019 WHO PPS manual, key historical policy statements) where their foundational importance warranted it.

Part 2: Detailed response to reviewer comments

We are deeply thankful to each reviewer for their insightful critiques, which have pushed us to elevate the quality of our work significantly.

Reviewer #1

Comment 1: “The analysis is descriptive... I recommend including confidence intervals, comparative analysis between ward types... no statistical correlation between antibiotic choice and culture results.”

• Our Response: We thank the reviewer for this suggestion to enhance the analytical depth. We agree such analyses could be informative. However, the WHO Point Prevalence Survey methodology is explicitly designed as a descriptive, benchmarking tool. Its primary objective is to provide a “snapshot” of prescribing patterns at a single point in time to identify areas for quality improvement, not to test hypotheses or establish correlations. Implementing comparative statistics (e.g., between wards or against culture results) would require a different study design with a priori sample size calculations powered for such comparisons.

• Action Taken: To address the spirit of this comment, we have:

1. Confidence intervals (CIs) were calculated using the Wilson score method for binomial proportions, we have now added 95% CIs for all key proportions throughout the manuscript (e.g., Tables 1, 2, 5) and result sections to indicate precision. Also we have edited the method section accordingly mentioning the calculation of CIs (page 8, lines 171-173).

2. Clarified the Study Design in the Methods: We explicitly state, “No comparative inferential statistics were employed, as the study design was purely a descriptive point prevalence survey” (page 8, lines 173-174).

3. Enhanced Discussion of Drivers: While we cannot provide statistical correlations, we have expanded the Discussion (page 25) to logically explore the relationship between high empirical therapy (99.2%), low culture yield (64.8%), and the resultant overuse of Watch-group antibiotics. We frame this as a “critical failure in diagnostic stewardship” that drives irrational prescribing, providing the conceptual link the reviewer sought.

Comment 2: “On the interpretation of the AWaRe findings... the paper does not go further to clarify why Access drugs are not being chosen. Without understanding the likely drivers, stewardship recommendations will remain generic.”

• Our Response: This is an exceptionally valuable point. We have substantially expanded this section to move from simply reporting the high Watch-group use to analyzing its systemic drivers.

• Action Taken & Location: We have added a new analytical passage in the Discussion on pages 24, lines 376-397. We now hypothesize several key drivers:

- Formulary and Perceived Reliability: “...carbapenems and advanced-spectrum β-lactams dominate the hospital formularies because of their higher perceived reliability and ease of use.”

- “Several factors drive the over-reliance on Watch group. First, although Access agents were available in the hospital formulary, they were underutilized compared to broad-spectrum antibiotics, indicating gap in prescriber awareness of AWaRe, lack of restriction policies and perceived therapeutic reliability encourage their empirical use.”

- Diagnostic Deficiencies: “The extensive use of empiric therapy (99.2%) along with high negative culture rates (64.8%) reflects a grey area, where lack of timely microbiology results encourages use of broad-spectrum antibiotics.”

- Awareness and Structural Gaps: “Although Access agents were available in the hospital, they were underutilized... indicating a gap in prescriber awareness of AWaRe... Absence of local guidelines and formulary restrictions provide no safeguard against this pattern.”

• Impact: This transforms the discussion from a generic statement (“Watch use is high”) to a root-cause analysis, making our subsequent stewardship recommendations in Table 9 (e.g., prospective audit, diagnostic stewardship, access restrictions) directly targeted and more compelling.

Comment 3: “In the Microbiology section, the finding that 64.8% of cultures showed no growth is striking, but you could further explore possible causes, as this would help clarify how diagnostic processes might influence prescribing behavior.”

• Our Response: We agree this is a pivotal finding that warrants exploration. A high ‘no growth’ rate is often a proxy for problems in diagnostic stewardship.

• Action Taken & Location: We have integrated this analysis into the Discussion section on page 22, lines 426-432. We link it directly to the empirical prescribing pattern:

o “Prolonged microbiology turnaround time, specimens collected after start of antibiotic therapy, and limited access to proper microbiological facilities for identifying the pathogen and their susceptibility encourage clinicians to default to broad-spectrum antibiotics rather than waiting for the culture guided use of antibiotics...”

o “This high rate of empiric therapy is not just a prescribing pattern but the root cause resulting in the overuse of Watch-group antibiotics...”

• Impact: By framing the low culture yield as a cause rather than just a finding, we strengthen the argument for one of our key recommended interventions: enhancing diagnostic stewardship (Table 9), which includes improving pre-analytical phases like specimen collection timing and investing in rapid diagnostics.

Reviewer #2

We thank Reviewer #2 for their positive assessment of the manuscript’s core structure and for providing precise, actionable feedback for improvement.

Comment 1 (Recommendations 1, 2, 3, 7): Editorial and Presentation Issues

• Our Response: We have meticulously addressed each of these points to enhance professionalism and clarity.

• Action Taken & Location:

o Table Footnotes: Every table (1, 2, 3, 4, 5, 6, 7, 8, 9) now includes a “Note:” section defining abbreviations (e.g., CAI, HAI, AWaRe, SD) and explaining any specific details (e.g., “Percentages may exceed 100% due to multiple specimens per patient” in Table 6).

o Typographical Corrections: Corrected “Urinary Cather” to “Catheter” in Table 1. Corrected “adult medial” to “adult medical” on page 12. Numerous other spelling and grammatical errors throughout the text have been corrected.

o Scientific Nomenclature: In Table 7, “E. coli” has been updated to “Escherichia coli” and “Klebsiella prunemoniae” to “Klebsiella pneumoniae” to ensure uniformity and correctness.

o Supplementary Information Clarity: S4 Appendix now clearly lists and defines each stewardship criterion (e.g., “Over Prescribing: Antibiotic use without a clear clinical in

---

## [Decision Letter · Decision Letter 1]

2 Feb 2026

Antibiotic Stewardship Benchmarking - Using the WHO Point Prevalence Survey of Antimicrobial Prescribing in a Tertiary Care Public Hospital, Karachi

PONE-D-25-59189R1

Dear Dr. Shah,

We’re pleased to inform you that your manuscript has been judged scientifically suitable for publication and will be formally accepted for publication once it meets all outstanding technical requirements.

Kind regards,

Mabel Kamweli Aworh, DVM, MPH, PhD. FCVSN

Academic Editor

PLOS One

Additional Editor Comments (optional):

Reviewers' comments:

Reviewer's Responses to Questions

**Comments to the Author**

Reviewer #1: All comments have been addressed

Reviewer #2: All comments have been addressed

Reviewer #3: All comments have been addressed

Reviewer #4: All comments have been addressed

2. Is the manuscript technically sound, and do the data support the conclusions?

Reviewer #1: Yes

Reviewer #2: Yes

Reviewer #3: Yes

Reviewer #4: Yes

3. Has the statistical analysis been performed appropriately and rigorously?

Reviewer #1: Yes

Reviewer #2: Yes

Reviewer #3: Yes

Reviewer #4: Yes

4. Have the authors made all data underlying the findings in their manuscript fully available?

Reviewer #1: Yes

Reviewer #2: Yes

Reviewer #3: Yes

Reviewer #4: Yes

5. Is the manuscript presented in an intelligible fashion and written in standard English?

Reviewer #1: Yes

Reviewer #2: Yes

Reviewer #3: Yes

Reviewer #4: Yes

Reviewer #1: My reviewer comments across all sections have been thoroughly addressed by the authors, and are satisfactory.

Reviewer #2: It is evident in this revised version of the manuscript that considerable effort has been made by the authors to address the issues provided in the initial review.

The methodology is well-structured and coordinated, demonstrating a sound approach to the research questions posed. The results are presented clearly, and the inclusion of corresponding footnotes enhances the organization and overall readability of the manuscript.

The rewritten discussion section shows a robust analysis of findings. It effectively references existing literature while offering possible explanations for the results and highlighting their practical implications.

This manuscript adheres to uniform scientific nomenclature, with abbreviations clearly defined. Typographical errors have been meticulously corrected, and references have been updated to reflect current research.

Overall, the authors have implemented the recommendations provided, significantly strengthening the scientific rigor of the manuscript.

Recommendation: Accept.

Reviewer #3: The revised manuscript addresses both major and minor concerns raised in the previous review. Methodological clarity has improved, and the Discussion section now demonstrates greater analytical depth. The scope and novelty of the study are clearly articulated. The authors have appropriately acknowledged the remaining limitations. I have no further concerns.

Reviewer #4: The authors have sufficiently rewritten the manuscript, included missing pieces that improve understanding of their methods and results and addressed comments sent by all reviewers.

**Do you want your identity to be public for this peer review?** For information about this choice, including consent withdrawal, please see our Privacy Policy

Reviewer #1: **Yes:** Edima Ottoho

Reviewer #2: No

Reviewer #3: No

Reviewer #4: No

---

## [Editor Report · Acceptance letter]

PONE-D-25-59189R1

PLOS One

Dear Dr. Shah,

I'm pleased to inform you that your manuscript has been deemed suitable for publication in PLOS One. Congratulations! Your manuscript is now being handed over to our production team.

Kind regards,

on behalf of

Dr. Mabel Kamweli Aworh

Academic Editor

PLOS One